# Loss of immune tolerance to IL-2 in type 1 diabetes

Louis Pérol[1,2,3,4,5,6,†], John M. Lindner[7], Pamela Caudana[4,5,6], Nicolas Gonzalo Nunez[4,5,6], Audrey Baeyens[1,2,3], Andrea Valle[8], Christine Sedlik[4,5,6], Delphine Loirat[5,6], Olivier Boyer[9,10,11], Alain Créange[12,13], José Laurent Cohen[14,15,16], Ute Christine Rogner[17], Jun Yamanouchi[18], Martine Marchant[7], Xavier Charles Leber[7], Meike Scharenberg[7], Marie-Claude Gagnerault[19,20,21], Roberto Mallone[19,20,21], Manuela Battaglia[8], Pere Santamaria[18,22], Agnès Hartemann[23,24], Elisabetta Traggiai[7] & Eliane Piaggio[1,2,3,4,5,6,†]

Type 1 diabetes (T1D) is characterized by a chronic, progressive autoimmune attack against pancreas-specific antigens, effecting the destruction of insulin-producing β-cells. Here we show interleukin-2 (IL-2) is a non-pancreatic autoimmune target in T1D. Anti-IL-2 autoantibodies, as well as T cells specific for a single orthologous epitope of IL-2, are present in the peripheral blood of non-obese diabetic (NOD) mice and patients with T1D. In NOD mice, the generation of anti-IL-2 autoantibodies is genetically determined and their titre increases with age and disease onset. In T1D patients, circulating IgG memory B cells specific for IL-2 or insulin are present at similar frequencies. Anti-IL-2 autoantibodies cloned from T1D patients demonstrate clonality, a high degree of somatic hypermutation and nanomolar affinities, indicating a germinal centre origin and underscoring the synergy between cognate autoreactive T and B cells leading to defective immune tolerance.

[1] Sorbonne Universités, Pierre and Marie Curie University Paris 06, Paris 75005, France. [2] Centre National de la Recherche Scientifique, UMR 7211, Paris 75013, France. [3] Institut National de la Santé et de la Recherche Médicale (INSERM), U 959, Immunology- Immunopathology-Immunotherapy (I3), Paris 75013, France. [4] Institut Curie, PSL Research University, INSERM U932, F-75005 Paris, France. [5] SiRIC TransImm ≪Translational Immunotherapy Team≫, Translational Research Department, Research Center, PSL Research University, Institut Curie, Paris F-75005, France. [6] Centre d'Investigation Clinique Biothérapie CICBT 1428, Institut Curie, Paris F-75005, France. [7] Novartis Institutes for Biomedical Research, Basel 4056, Switzerland. [8] Diabetes Research Institute (DRI), IRCCS San Raffaele Scientific Institute, Milan 20132, Italy. [9] INSERM, U905, Rouen 76183, France. [10] Normandie Univ. IRIB, Rouen 76183, France. [11] Rouen University Hospital, Laboratory of Immunology, Rouen 76183, France. [12] Service de Neurologie, Groupe Hospitalier Henri Mondor, AP-HP, Créteil F-94010, France. [13] EA 4391, Université Paris Est, Créteil F-94010, France. [14] Université Paris-Est Créteil, Créteil F-94010, France. [15] INSERM U 955, Institut Mondor de Recherche Biomédicale (IMRB), Créteil F-94010, France. [16] AP-HP, Groupe Hospitalier Henri-Mondor Albert-Chenevier, CIC-BT-504, Créteil F-94010, France. [17] Institut Pasteur, CNRS URA 2578, Département Biologie du développement et cellules souches, Paris 75015, France. [18] Julia McFarlane Diabetes Research Centre and Department of Microbiology, Immunology and Infectious Diseases, Snyder Institute for Chronic Diseases, Cumming School of Medicine. University of Calgary, Calgary, Alberta, Canada T2N 4N1. [19] INSERM, U1016, Cochin Institute, DeAR Lab, Paris 75014, France. [20] Assistance Publique-Hôpitaux de Paris, Hôpital Cochin, Service de Diabétologie, Paris 75014, France. [21] Paris Descartes University, Sorbonne Paris Cité, Faculté de Médecine, Paris 75270, France. [22] Institut D'Investigacions Biomediques August Pi i Sunyer, Barcelona 08036, Spain. [23] Department of Medicine Faculty, Université Pierre et Marie Curie—Paris 6, Paris 75005, France. [24] Department of Endocrinology, Nutrition and Diabetes, Assistance Publique-Hôpitaux de Paris (AP-HP), Pitié-Salpêtrière-Charles Foix Hospital, Paris 75013, France. † Present address: INSERM U932, 26 rue d'Ulm, Paris 75005, France. Correspondence and requests for materials should be addressed to E.P. (email: eliane.piaggio@curie.fr).

Anti-cytokine antibodies have been reported in healthy individuals as well as in patients with infectious and autoimmune diseases, for example, anti-interferon (IFN)-γ antibodies in mycobacterial infections, anti-granulocyte-macrophage colony-stimulating factor (GM-CSF) antibodies in severe autoimmune pulmonary alveolar proteinosis, and anti-interleukin (IL)-17 antibodies in mucocutaneous candidiasis[1,2]. However, the stimuli eliciting anti-cytokine antibody responses, and whether these antibodies are pathologically causative *in vivo*, remains unknown. IL-2 is pleiotropic but indispensable for proper $T_{reg}$ cell function[3], making it an attractive target in autoimmune disease. In type 1 diabetes (T1D), circulating, pancreas-specific autoantibodies are correlated with the onset of a chronic, pancreatic autoimmune attack leading to the progressive loss of insulin-producing β-cells[4]. Defects in the induction of central and peripheral tolerance checkpoints[5] are notable contributors to T1D pathology. Illustrating this point, non-obese diabetic (NOD) mice, which recapitulate many characteristics of the complex pathogenesis of human T1D[6], and T1D patients develop both islet-specific autoantibodies and autoreactive T cells, and feature syntenic genetic linkage to disease[7].

## Results

We have previously shown that administration of low doses of recombinant human IL-2 (rhIL-2) at onset reverts disease in about half of the treated NOD mice[8]. To analyse if doses 10-, 20-, or 40-fold higher than what we have previously shown to be effective to revert hyperglycaemia in new onset diabetic NOD mice could increase treatment efficacy, we administered $2.5 \times 10^5$, $5 \times 10^5$, or $10^6$ IU rhIL-2 to pre-diabetic NOD mice[9]. We observed that these higher rhIL-2 doses were, in a dose-dependent manner: (i) lethally toxic in half of the mice; (ii) precipitated diabetes onset in around 25% of them; or intriguingly, (iii) induced no apparent clinical signs in around 25% of the mice, even after a 30-day administration (Fig. 1a–c). Interestingly, after 5 days of treatment, mice responded to all doses of administered rhIL-2 by increasing $T_{reg}$ cell frequencies, which returned to pre-treatment levels by day 30 after IL-2 administration (Fig. 1d). We reasoned that mice surviving 30 days post-high-dose rhIL-2 treatment may have developed antibodies capable of neutralizing the injected rhIL-2. Indeed, only sera from the surviving rhIL-2-treated NOD mice demonstrated high titres of rhIL-2 immunoglobulin-γ IgG as detected by enzyme-linked immunosorbent assay (ELISA) (Fig. 1e). Furthermore, those sera efficiently neutralized rhIL-2 biological activity in an *in vitro* assay using IL-2-dependent CTLL-2 cells (Fig. 1f), suggesting that they were responsible for the *in vivo* resistance to the side effects of high rhIL-2 doses. Interestingly, sera from untreated NOD mice, but not autoimmunity-resistant B6 mice, also showed detectable anti-rhIL-2 antibodies (Fig. 1e). These results suggested the existence of pre-formed antibodies capable of binding to rhIL-2, possibly representing naturally occurring, cross-reactive autoantibodies against murine IL-2. Indeed, only sera from untreated pre-diabetic and diabetic NOD mice, but not from B6 and BALB/c, reacted to mIL-2. Notably, IgG anti-mIL-2 autoantibody titres were significantly higher in overtly diabetic NOD mice as compared to their pre-diabetic counterparts (Fig. 2a). In NOD mice, IgG anti-mIL-2 autoantibodies were mostly of the IgG2b subclass (Fig. 2b). We confirmed the specificity of anti-mIL-2 autoantibodies by competitive binding to IL-2-coated beads (Supplementary Fig. 1a,b, Supplementary Methods), and observed that anti-mIL-2 autoantibodies showed *in vitro* neutralizing activity, inhibiting CTLL-2 cell growth in a dose-dependent manner (Fig. 2c). Interestingly, anti-mIL-2

autoantibody titres increase with age and, consequently, with T1D progression (Fig. 2a,d). Moreover, NOD females generate higher titres of anti-mIL-2 autoantibodies than males of the same age, correlating with the higher frequency of spontaneous T1D incidence in females (Fig. 2e).

It has been shown that anti-insulin autoantibody (IAA) titres in 8-week-old mice are predictive of diabetes onset[10]. Strikingly, anti-mIL-2 autoantibody titres display a similar positive correlation with shorter time to T1D onset at just 6 weeks after birth (Fig. 2f). Thus, the spontaneously produced anti-IL-2 autoantibodies in young NOD mice could be used for predicting diabetes well before disease onset, either alone or in combination with IAA. In contrast to IAA titres, which can oscillate during disease progression in NOD mice[11], anti-mIL-2 autoantibody titres appear at a stable trajectory preceding or concomitant with disease progression. This may be due to the cyclical appearance of insulin as an autoantigen during waves of pancreatic destruction, whereas IL-2 is more persistently present to drive a response.

T1D susceptibility and resistance alleles on mouse chromosome 3 (*Idd3*) correlate with differential expression of IL-2 (ref. 12). NOD mice carrying the *Idd3* locus from B6 mice (NOD.*Idd3*[B6]) are resistant to T1D development and their T cells produce two-fold more IL-2 than NOD mice. Conversely, IL-2-haploinsufficient NOD mice (NOD.IL-2$^{+/-}$) produce half as much IL-2 as NOD mice and have accelerated diabetes[12]. To study the effect of fluctuation in constitutive IL-2 production levels on the generation of anti-mIL-2 autoantibodies, we quantified anti-mIL-2 autoantibodies in the sera of these different NOD congenic strains. Female NOD.IL-2$^{+/-}$ mice had higher anti-mIL-2 autoantibody titres than NOD mice, and NOD.*Idd3*[B6] mice had the lowest levels (Fig. 2g). As controls for disease resistance independent of the *Il2* locus, we used NOR mice, which represent a major histocompatibility complex-matched diabetes-resistant control strain for NOD mice that share the *Idd3*[NOD] locus, but carry *Idd5.2*, *Idd9/11* and *Idd13* B6 protective loci, and NOD.*Idd6*[C3H] congenic mice, which share the *Idd3* locus but are less susceptible to T1D development[13]. In these two strains, although insulitis and diabetes are reduced or absent, anti-mIL-2 autoantibodies are present, indicating that, while their presence is associated with T1D development, they are not sufficient to induce T1D. The different congenic strains produce different amounts of IL-2 that could be complexed with circulating anti-mIL-2 autoantibodies and therefore undetected by a typical ELISA, which only detects free anti-mIL-2 autoantibody. The levels of free anti-mIL-2 autoantibody and IL-2/anti-mIL-2-autoantibody immune complexes follow similar distributions across the congenic strains (Fig. 2h), confirming that increased genetically determined levels of IL-2 correlate with lower anti-mIL-2 autoantibody production. These data support the hypothesis that reduced functional IL-2 levels are responsible for broken self-tolerance. Along these lines, injection of anti-mIL-2-autoantibody-depleted versus undepleted serum in NOD mice induced a reduction of CD25 expression on blood $T_{reg}$ cells (similar to the injection of equivalent amounts of a control anti-IL-2 neutralizing antibody (S4B6), suggesting that anti-mIL-2 autoantibodies could contribute to reduced IL-2 availability and impact $T_{reg}$ fitness *in vivo* (Supplementary Fig. 2, Supplementary Methods). Interestingly, while IL-2-deficient mice are defective for thymic formation of a subset of islet-specific $T_{reg}$ cells[14], we do not observe a correlation between Foxp3-positive T-cell numbers and anti-mIL-2 autoantibodies in NOD mice, suggesting this effect is peripheral, and not thymus-driven.

Long-lasting serum antibody production is mainly driven by long-lived plasma cells located in bone marrow and memory B cells resident in secondary lymphoid organs[15]. Indeed, the bone marrow of NOD, but not B6 mice, contained anti-IL2 IgG

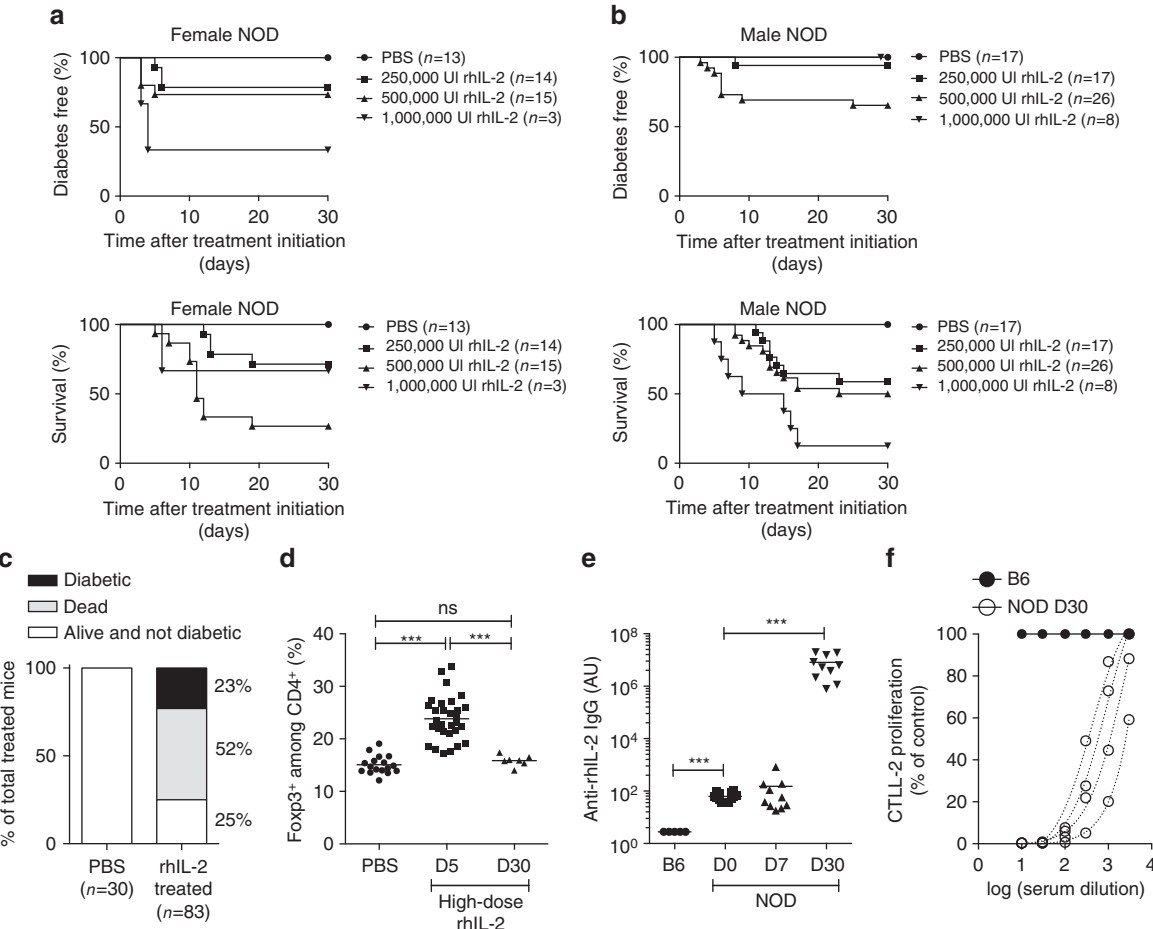

**Figure 1 | High-doses rhIL-2 injection in NOD induce neutralizing anti-rhIL-2 antibodies.** (a–f) Five-to-14-week-old male or female NOD mice were daily treated with PBS or high-doses rhIL-2 (250,000; 500,000 or 1,000,000 IU) over 30 days. (**a,b**) Kaplan-Meier survival curves of treated female (**a**, top panel) or male (**b**, top panel) mice; and diabetes incidence in female (**a**, bottom panel) or male (**b**, bottom panel) mice. (**c**) Percentage of dead, diabetic or alive and non-diabetic NOD mice after 30 days of treatment; IL-2-treated: pool of (250,000; 500,000 and 1,000,000 IU IL-2 treated mice. (**d**) Percentage of Foxp3$^+$ among CD3$^+$ CD4$^+$ splenocytes of NOD mice treated for 5 to 30 days with high-doses IL-2 or PBS. (**e**) Serum anti-rhIL-2 IgG titres of untreated B6 mice and pre-diabetic NOD mice treated for 0, 7 or 30 days with high-dose IL-2. (**f**) Proliferation of CTLL-2 cells cultured for 3 days with 3 IU ml$^{-1}$ rhIL-2 and serially diluted serum from B6 (closed circles) or NOD mice treated for 30 days with high-dose rhIL-2 (open circles). Proliferation is expressed as percentage of control (CTLL-2 cultured for 3 days with 3 IU ml$^{-1}$ rhIL-2 without mouse serum). Data are cumulative of at least two independent experiments. ns, not significant. ***$P < 0.001$ (non-parametric Mann-Whitney test).

secreting cells (Fig. 3a), and mIL-2-specific memory B cells were present in the spleen of most NOD mice (Fig. 3b).

To evaluate the presence of IL-2-reactive T cells, we generated a library of 53 peptides (15-mers with 12 aa overlap, Supplementary Table 1) derived from mIL-2 and measured IFN-γ production by B6 and NOD splenocytes upon *in vitro* stimulation. No IFN-γ production was detected from B6 splenocytes (Fig. 3c), but cells from NOD mice specifically recognized two mIL-2-derived peptides (in bold in Fig. 3d). The mIL-2$_{10-24}$ epitope spans the signal peptide and may thus be preferentially processed and presented by the major histocompatibility complex Class I pathway, as described for the preproinsulin signal peptide in T1D[16]. The other peptide recognized, mIL-2$_{67-81}$, likely represents an immunodominant target of IL-2 autoreactivity. As a control, we included the class II-restricted BDC2.5 mimotopic peptide (P31)[17] that induced IFN-γ production only in NOD mice and to similar levels as those induced by the IL-2-derived peptides. These experiments establish the existence of IL-2-specific auto-reactive B and T cells in NOD mice.

Extrapolating the discovery of IL-2 as a novel autoantigen in mouse models of T1D to humans, we found that a significantly higher fraction of T1D patient sera (from three independent cohorts) contain anti-rhIL-2 autoantibodies than either healthy donors or type 2 diabetic patients (who present chronic hyperglycaemia in the absence of islet autoimmunity) (Fig. 4a and Supplementary Tables 2 and 3). Interestingly, as in the NOD mice, there was a positive correlation between anti-rhIL-2 autoantibody titres and patient age (Supplementary Fig. 3). The reduced penetrance of anti-rhIL-2 autoantibodies in T1D patients relative to the NOD mice may be attributed to the genetic and temporal heterogeneity of the human disease, such that only a subset of diabetic patients are phenotypically similar to the NOD model with respect to anti-IL-2 autoantibodies. To resolve the composition of the anti-IL-2 IgG autoantibodies, we analysed the IgG subclasses in the sera of T1D patients for anti-IL-2 specificity, and found that most samples contain IgG1 and IgG2 anti-rhIL-2 autoantibodies (Fig. 4b). To estimate the EC$_{50}$ of the anti-rhIL-2 autoantibody, we purified IgG from the sera of T1D patients. Purified IgG samples with higher affinity for IL-2 showed a

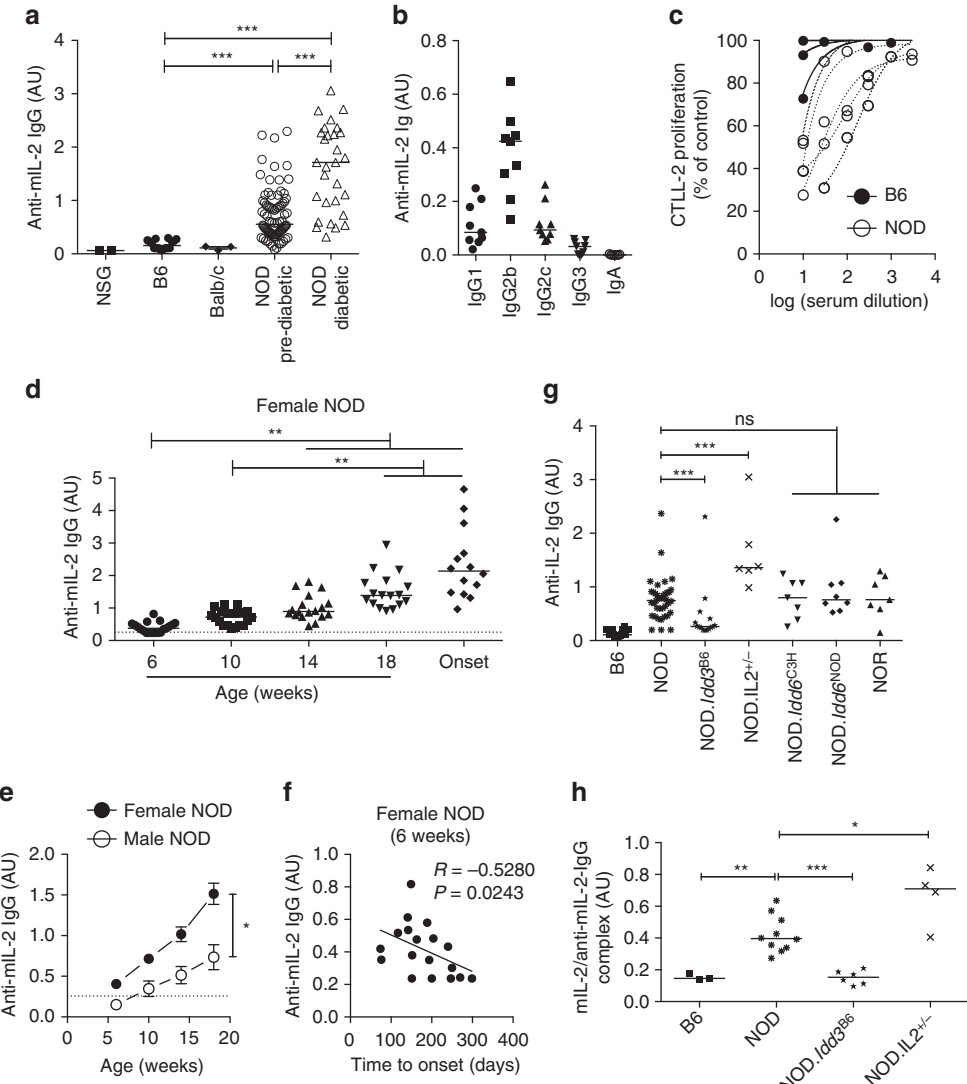

**Figure 2 | Anti-mIL-2-autoantibodies in NOD mice. (a–c)** Serum samples were obtained from different mouse strains: NSG (NOD *scid* gamma), B6, Balb/c, pre-diabetic NOD (NOD Pre-diabetic) and diabetic NOD (NOD Diabetic). **(a,b)** Serum titres of anti-mIL-2 IgG **(a)**, IgG isotypes (IgG1, 2b, 2c and 3) and IgA **(b)** in the different strains. **(c)** Proliferation of CTLL-2 cells cultured for 3 days with 1 ng ml$^{-1}$ mIL-2 and different concentrations of B6 (closed circles) or NOD (open circles) sera. Proliferation is expressed as percentage of control (CTLL-2 cultured for 3 days with 1 ng ml$^{-1}$ mIL-2 without mouse serum, mean c.p.m. of 84,590). **(d–f)** Sera were obtained at different ages after birth and at disease onset (Onset) in two independent cohorts of female NOD mice ($n = 13$ and 6, respectively) and one cohort of male NOD mice ($n = 5$). **(d,e)** Serum anti-mIL-2 IgG titres in NOD mice in function of the age **(d)** or of the sex **(e)**. Dashed line indicates the mean value given by B6 mouse sera in the corresponding ELISA. **(f)** Correlation between anti-mIL-2 IgG titres at 6 weeks and time to onset of diabetes in female NOD mice (non-parametric Spearman correlation test). **(g,h)** Serum samples were obtained from different mouse strains (all females and age-matched): B6, wild-type NOD, NOD.*Idd3*[B6], *Il2*-hemizygous NOD: NOD.*Idd3*[NOD/NOD-IL-2null] (NOD.*Il2*$^{+/-}$), NOD.*Idd6*[C3H] and their corresponding controls NOD.*Idd6*[NOD], as well as NOR. Serum titres of anti-mIL-2 IgG **(g)** and IL-2/anti-rhIL-2-autoantibodies complex **(h)** in the different mouse strains. Symbols and curves represent individual mice, horizontal bars are the medians and error bars represent the s.e.m. Data are cumulative of at least two independent experiments. ns, not significant. *$P < 0.05$; **$P < 0.01$; ***$P < 0.001$ (non-parametric Mann-Whitney test).

significant correlation to high serum titres (Fig. 4c). The specificity of the anti-rhIL-2 autoantibodies from all anti-rhIL-2-autoantibodies$^+$ samples was confirmed by competition ELISA (Fig. 4d).

To understand the ontogeny of anti-hIL-2 autoantibodies in type 1 diabetics, we isolated circulating human IgG memory B cells from four T1D patients and clonally expanded them *in vitro* at very low density (1–4 cells/well) to drive their differentiation into antibody-secreting cells[18]. After 12 days, we analysed culture supernatants for the presence of anti-influenza, anti-insulin, and anti-IL-2 IgG antibodies. Similar frequencies of autoreactive anti-insulin and anti-IL-2 IgG memory B cells were

present (albeit at lower frequencies than for, for example, influenza virus as a stereotypical recall antigen), indicating the existence of a specific, persistent immune response against IL-2 in T1D (Fig. 5a). Heavy and light chain sequences of anti-IL-2-specific BCRs (B cell receptor) were cloned from three independently expanded B cells from a single patient. Sequence analysis revealed a clonal origin ($V_H3$–30/$V_K1$–39) of all three hits, as well as extensive somatic hypermutation from the germline sequence (Fig. 5b). Recombinantly produced antibodies were found to have an affinity for rhIL-2 between 6 and 32 nM (Fig. 5b). Taken together, this evidence indicates

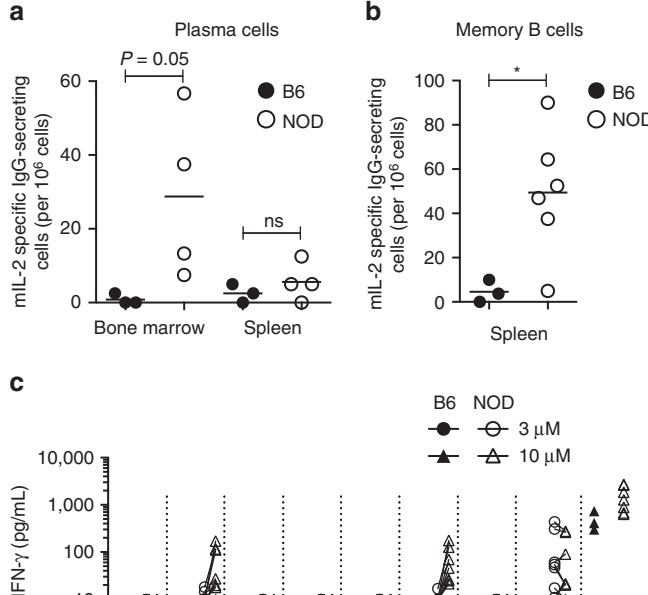

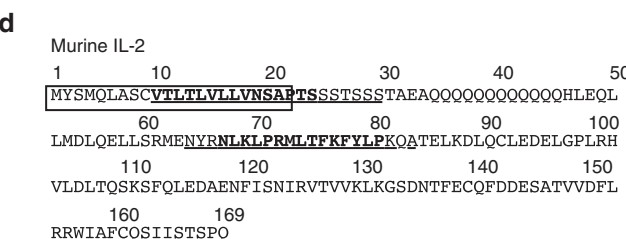

**Figure 3 | NOD mice present B and T autoimmune responses against mIL-2.** (**a**,**b**) The number of anti-mIL-2 secreting cells in fresh splenocytes and bone marrow cells (plasma cells, **a**) or in 5-days CpG pre-activated splenocytes (memory B cells, **b**) from 10–18 week-old female B6 or pre-diabetic NOD mice was quantified by ELISPOT. (**c**) IFN-γ production by 10–18-week-old female B6 ($n = 3$) or pre-diabetic NOD ($n = 7$) splenocytes was quantified in culture supernatants by CBA after 72 h of stimulation with DMSO, mIL-2 peptides that gave a positive response in the initial screen (3 and 10 μM of each peptide), P31 peptide (3 and 10 μM) or aCD3-CD28 coated beads (ratio 1bead:1cell). (**d**) mIL-2 amino acid sequence. The regions corresponding to peptides that gave positive results in the initial screen are underlined and the immunodominant peptides (mIL-2$_{10-24}$ and mIL-2$_{67-81}$) are in bold. The mIL-2 signal peptide is indicated by a rectangle. (**a–c**) Symbols represent individual mice and horizontal bars are the medians. Data are cumulative of at least two independent experiments. (**a**,**b**) *$P < 0.05$ (Kruskal-Wallis test with Dunn's multiple comparisons test).

an affinity matured, germinal centre origin for anti-rhIL-2 autoantibodies.

To assess T-cell reactivity to rhIL-2 by IFN-γ ELISPOT following *in vitro* antigen stimulation[19], we generated a library of 30 peptides covering the entire human IL-2 sequence and included four peptides covering regions comprising the two mutations introduced in therapeutic recombinant rhIL-2 (Proleukin) relative to native IL-2 (hIL-2) (Supplementary Table 4). Relative to age-matched healthy donors, a higher frequency of T1D patients displays T-cell reactivity against hIL-2, with two significantly enriched peptides (in bold in Fig. 5c). The T1D-associated intracellular insulinoma-associated antigen 2

(IA-2) included as a control was targeted by T cells at the same frequency in T1D patients (38%) as the hIL-2$_{56-70}$ peptide; of the five T1D patients responding to IA-2, three also responded to hIL-2$_{56-70}$. Intriguingly, this hIL-2$_{56-70}$ peptide (LTRMLTFKFYMPKKA) overlaps with the orthologous mIL-2$_{67-81}$ (NLKLPRMLTFKFYLP) peptide found targeted in NOD mice (Supplementary Fig. 4), suggesting that this epitope is immunodominant across species.

This concerted dual T- and B-cell adaptive autoimmune response strongly supports the hypothesis that tolerance to IL-2 is broken in T1D, demonstrating uniquely that autoimmunity is also directed against non-islet antigens in T1D. The physio-pathology of T1D has previously been linked to genetic defects in the IL-2/IL-2R pathway that indirectly affect immune tolerance, partly by disrupting T$_{reg}$ cell function[12,20,21]. Adaptive autoimmune responses against IL-2 add another piece to this puzzle, further reinforcing the link between impaired IL-2 bioavailability and T1D. Interestingly, significantly increased percentages of sera from rheumatoid arthritis (RA), Sjögren syndrome, systemic lupus erythematosus and autoimmune myositis patients were also anti-rhIL-2-autoantibodies+, and anti-mIL-2 autoantibodies were found in lupus-prone mice (Fig. 6); indicating that IL-2 autoreactivity plays a previously unsuspected role in the pathogenesis of different autoimmune conditions in human and mouse.

## Discussion

Steady-state levels of IL-2 are too low to stimulate effector T and NK cells but are critical for the maintenance of T$_{reg}$ cells[3]. Thus, a reduction in peripheral IL-2 bioavailability may interfere with T$_{reg}$ homeostasis and function, and lead to broken immune tolerance. This effect is well illustrated by the administration of neutralizing anti-IL-2 antibodies, which triggers autoimmune gastritis in BALB/c mice and accelerates T1D in NOD mice. The effect in NOD mice is accompanied by numerous autoimmune presentations, including gastritis, thyroiditis, sialadenitis, and neuropathy. Of note, short-term low-dose IL-2 administration induces variable T$_{reg}$ cell expansion in T1D[22,23] and autoimmune vasculitis patients[24]. Thus, it would be interesting to verify whether the *de novo* presence or treatment-induced production of anti-rhIL-2 autoantibodies underlies these variable responses.

While the highly significant association between free soluble anti-rhIL-2 autoantibodies and T1D is remarkable, the functional capacity of these antibodies—at the monoclonal level—to neutralize or otherwise modulate IL-2 function remains to be assessed. Such antibodies could be useful diagnostic, experimental or therapeutic agents in the field of autoimmune disease. Intriguingly, the presence of low levels of IL-2-complexed anti-rhIL-2 IgG in healthy individuals has been described some time ago[25]; however, the origin and function of these Ab have not been addressed in the years since. In light of our results, it will be especially interesting to revisit this topic and determine the prevalence of these autoantibodies in a healthy population, as well as whether their functionality differentiates them from T1D-derived anti-rhIL-2 autoantibodies. With the proper tools now in hand to characterize the BCR repertoire underlying these autoreactive antibodies, we can return to long-standing questions with a modern perspective and insight into clinically relevant aspects of human health.

## Methods

**Mice and diagnosis of diabetes.** C57BL/6 (B6) and Balb/c mice were obtained from Janvier or Charles River Laboratories. NOD.Cg-*Prkdc*$^{scid}$ *Il2rg*$^{tm1Wjl}$/SzJ (NOD *scid* gamma, NSG) and NOD mice were bred in our animal facility under specific pathogen-free conditions. NOD.*Idd3*$^{B6}$, NOD.*Idd3*$^{NOD/NOD-IL-2null}$ (NOD.*Il2*$^{+/-}$), NOD.*Idd6*$^{C3H}$, NOD.*Idd6*$^{NOD}$, NOR and B6/*lpr* strains were

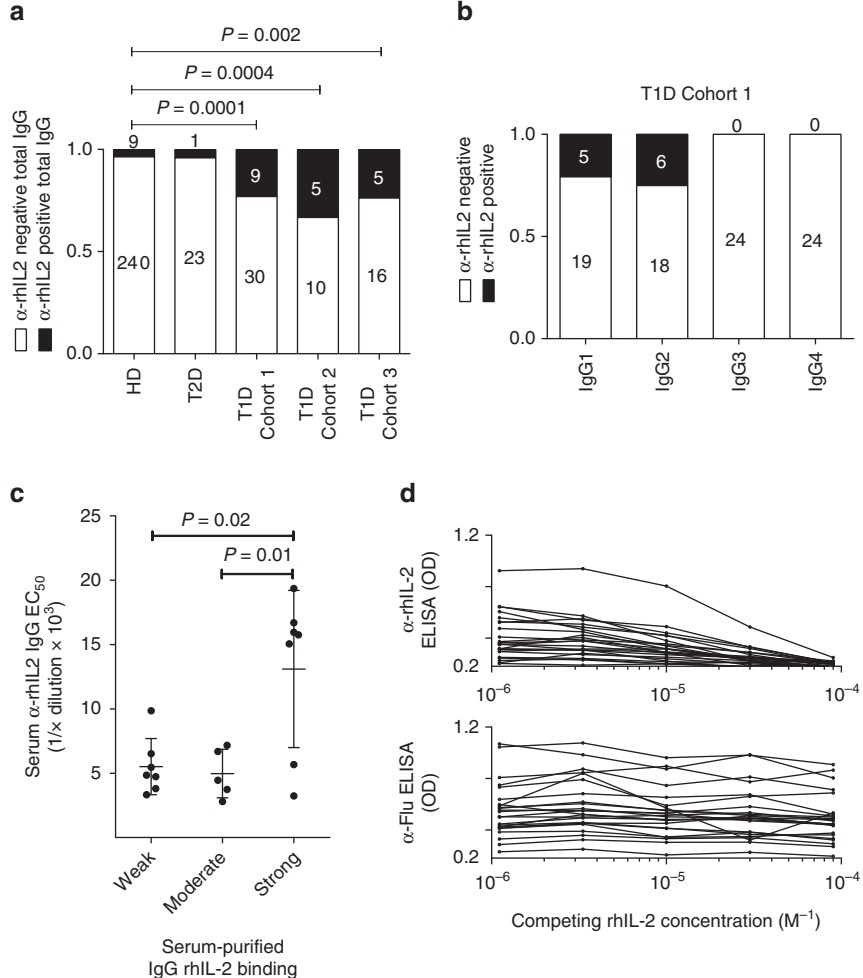

**Figure 4 | T1D patients present a humoural autoimmune response against IL-2.** (**a**) Percentage of anti-rhIL-2 positive subjects among five different cohorts: serum samples were obtained from healthy donors (HD, $n = 249$) and patients diagnosed with type 2 (T2D; $n = 24$) or type 1 diabetes ($n = 39$ in cohort 1, $n = 15$ in cohort 2 and $n = 21$ in cohort 3). P-values were calculated using pairwise Fisher exact tests. (**b**) IgG subclass-specific anti-IL2 ELISA in T1D patients, absorbance OD measured at 450 after incubation of sera diluted 1:50. (**c**) T1D patient samples were divided into three groups based on $EC_{50}$ values for an IL-2 direct ELISA from total sera (weak, moderate and strong binders) and are plotted relative to the anti-IL-2 $EC_{50}$ value of IgG purified from the respective serum sample. Mean and s.d. overlay individual data points, and P-values were calculated using Welch's t-test. (**d**) (upper panel) rhIL-2 competition ELISA using IgG purified from the sera of T1D patients, with the control (lower panel) of an anti-influenza ELISA with competing soluble rhIL-2.

described previously[12,13,26]. For diabetes follow-up, glycosuria was measured every 2 days using colorimetric strips (Multistix, Bayer) and glycaemia was quantified by a glucometer (Optium Xceed, Abbott Diabetes Care). Diabetes onset was defined by two consecutive positive measures of glycosuria and confirmed by hyperglycaemia ($> 250\,\mathrm{mg\,dl}^{-1}$). Manipulations were performed according to European Union guidelines and with approval by The Ethics Committee in Animal Experiment Charles Darwin, France (no. Ce5/2012/021).

**rhIL-2 treatment.** Five- to fourteen-week-old female and male NOD mice were treated with daily intraperitoneal injections of 250,000; 500,000 or 1,000,000 IU of rhIL-2 (Proleukin, Novartis) for 30 consecutive days.

**Flow cytometry.** Blood was collected by retro-orbital puncture. Spleen cells were obtained by mechanical disruption. After Fc receptor blocking (clone 2.4G2, 1:1,000; BD Biosciences), cells were stained with anti-CD4 PerCP-Cy5.5 (clone RM4-5; 1:800; BD Biosciences) or CD4 BV785 (clone RM4-5; 1:800; BioLegend), anti-TCR-β APC (clone H57-597; 1:200; BD Biosciences) or PE (clone H57-597; 1:100; BD Biosciences), anti-NKp46 FITC (clone 29A1.4; 1:300; eBiosciences), anti-CD8 Alexa700 (clone 53-6.7; 1:1,600; BioLegend), anti-CD25 PE-Cy7 (clone PC61; 1:300; eBiosciences), anti-CD62L BV605 (clone MEL-14; 1:800; BioLegend) and anti-CD44 APC-Cy7 (clone IM7; 1:400; BD Biosciences) in PBS/3% FCS. Then anti-Foxp3 eFluor450 (clone FJK-16 s; 1:150; eBioscience) and Ki67-PE (clone B56; 1:50; BD Biosciences) or Ki67-Alexa647 (clone B56; 1:50; BD Biosciences) staining was performed using the eBioscience

Foxp3 staining kit according to the manufacturer's instructions. Cells were acquired on a Fortessa flow cytometer and analysed with FlowJo software (TreeStar).

**Quantification of anti-IL-2 human and mouse antibodies.** Serum titres of anti-mIL-2 autoantibodies or anti-rhIL-2 autoantibodies were quantified by ELISA. Microtitre 96-well plates (Medisorp, Nunc) were incubated overnight at 4 °C with 100 μl per well of carbonate coating buffer (pH 9.6) containing $1\,\mathrm{\mu g\,ml}^{-1}$ mIL-2 (PeproTech) for the detection of anti-mIL-2 autoantibodies, or with 100 μl per well of $10^5\,\mathrm{IU\,ml}^{-1}$ rhIL-2 (Proleukin, Novartis) for the detection of anti-rhIL-2 autoantibodies. After blocking with PBS/2% bovine serum albumin (BSA) for 2 h, plates were incubated with 50 μl of serially diluted serum samples in duplicate for 2 h at room temperature. After extensive washing with PBS/0.1% Tween-20, biotin-labelled anti-mouse IgG, IgG1, IgG2b, IgG2c, IgG3, IgM or IgA (all 1:5000; Southern Biotech) was added to each well and the plates were kept at room temperature for 1 h. Plates were subsequently incubated with horseradish perox-idase (HRP)-conjugated streptavidin (1:2000; Invitrogen) for 30 min followed by 3,3′,5,5′-tetramethylbenzidine (TMB) substrate (eBioscience or BD Biosciences) for 10 min. The reaction was acid blocked and absorbance was read at 450 nm with a DTX 880 Multimode Detector (Beckman Coulter). A standard curve was calculated using two-fold serial dilutions of rat anti-mouse IL-2 (for anti-mIL-2 auto-antibodies, clone JES6-1A12, eBioscience) or of rat anti-human IL-2 (for anti-rhIL-2 autoantibodies, clone MQ1-17H12, eBioscience) revealed with an HRP-con-jugated goat anti-rat Ig (1:2000; Dako).

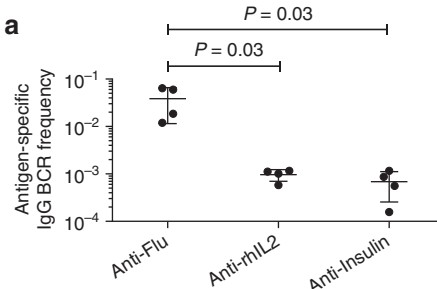

| Patient ID | N16 | N22 | N32 | N43 |
|---|---|---|---|---|
| Total BCRs | 10,326 | 3,589 | 3,442 | 12,767 |
| Anti-Flu | 192 | 230 | 206 | 153 |
| Anti-rhIL2 | 12 | 4 | 2 | 13 |
| Anti-Insulin | 12 | 2 | 3 | 2 |

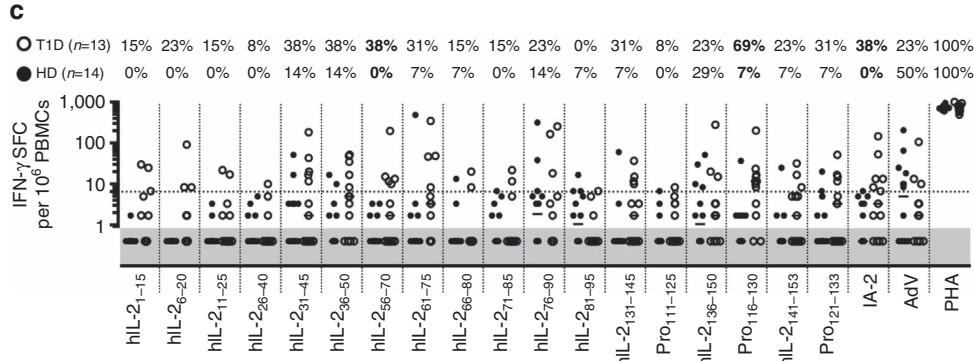

| | | | | Nucleotide mutations from germline sequence | | | | | CDR3 amino acid sequence | SET IC$_{50}$ (nM) |
|---|---|---|---|---|---|---|---|---|---|---|
| | | | | FR1 | CDR1 | FR2 | CDR2 | FR3 | | |
| N43-1I5 | Heavy chain | V$_H$3–30 | J$_H$6 | 9 | 9 | 6 | 8 | 24 | ANRMVNYYGMDV | 32 ± 18 |
| | Light chain | V$_K$1–39 | J$_K$3 | 4 | 5 | 6 | 8 | 9 | QESFVTRLFT | |
| N43-2N10 | Heavy chain | V$_H$3–30 | J$_H$6 | 9 | 7 | 5 | 8 | 19 | ANRLVNYYGMDV | 16 ± 11 |
| | Light chain | V$_K$1–39 | J$_K$3 | 5 | 4 | 6 | 11 | 6 | QESYVTRQFT | |
| N43-5F21 | Heavy chain | V$_H$3–30 | J$_H$6 | 10 | 7 | 7 | 8 | 20 | ANRLVNYYGMDV | 6 ± 4 |
| | Light chain | V$_K$1–39 | J$_K$3 | 1 | 6 | 1 | 2 | 5 | QESYVIRLFT | |

**Figure 5 | Circulating anti-IL-2-specific B and T lymphocytes in T1D patients.** (**a**) (Left) Relative frequency and (right) absolute numbers of antigen-specific IgG memory B cells across four T1D patients. Error bars represent s.d., $P$-values calculated using an unpaired $t$-test. (**b**) IL-2-specific heavy and light chain V gene segment usage, CDR3 amino acid sequences, and rhIL-2 solution equilibrium titration (SET) affinity for recombinant IgG obtained from a single T1D patient, (**c**) IFN-$\gamma$ production by peripheral blood mononuclear cells (PBMCs) from healthy donors (HD) ($n = 14$, closed circles) or T1D patients ($n = 13$, open circles) quantified by ELISPOT after stimulation with rhIL-2 (Proleukin, Pro) or Pro peptides (10 µM per each), intracellular IA-2, adenovirus lysate (AdV), or PHA. The number of IFN-$\gamma$ spot-forming cells (SFC)/10$^6$ PBMCs is depicted, the dashed line indicates the positive cut-off threshold, and the grey shaded area shows undetectable responses (that is, identical to spontaneous background responses; see material and methods for threshold determination). The percent of positive T1D (top number) and HD (bottom number) is indicated for each condition, with antigens yielding responses significantly different between HD and T1D patients in bold ($P < 0.03$ using the Fisher exact test).

**Quantification of IL-2/anti-mIL-2 immune complexes in sera.** Mice were bled from the retro-orbital sinus, and serum titres of IL-2/anti-mIL-2-autoantibodies immune complexes quantified by ELISA. Microtitre 96-well plates (Medisorp, Nunc) were incubated overnight at 4 °C with 100 µl per well of carbonate coating buffer (pH 9.6) containing 0.5 µg ml$^{-1}$ polyclonal anti-mIL-2 (PeproTech). After blocking with PBS/2% BSA for 2 h, plates were incubated with 50 µl of serially diluted sera in duplicate for 2 h at room temperature. After extensive washing with PBS/0.1% Tween20, biotin-labelled anti-mouse IgG (1:5000; Southern Biotech) was added to each well and the plates were kept at room temperature for 1 h. Plates were subsequently incubated with HRP-conjugated streptavidin and TMB substrate as above and read before.

**CTLL-2-based neutralization assays.** CTLL-2 cells (ATCC, mycoplasma-free) were cultured (10$^4$ cells per well) in 96-well plates in complete Roswell Park Memorial Institute (RPMI) medium (Gibco) containing no mIL-2, 1 ng ml$^{-1}$ mIL-2 or 3 IU ml$^{-1}$ rhIL-2 with or without heat-inactivated (30 min at 56 °C) serially diluted serum from B6 or NOD mice. After 48 h, cultures were pulsed 18 h with [$^3$H]-thymidine (1 µCi per well) and counted by liquid scintillation.

**B-cell ELISPOT.** After activation with 35% ethanol, 96-well polyvinylidene difluoride plates (MAIP4510, Millipore) were coated with 70 µl per well of 5 µg ml$^{-1}$ mIL-2 (Peprotech) overnight at 4 °C. After washing with PBS,

plates were blocked with Protein-Free Blocking Buffer (Thermo) for 1 h at room temperature and then with complete RPMI medium for 30 min at room temperature. Serially diluted spleen or bone marrow cells (5 × 10$^4$ to 4 × 10$^5$ cells per well in complete RPMI medium) from 10- to 18-week-old female B6 or NOD mice were added in the ELISPOT plate. In a set of experiments, splenocytes from 10- to 18-week-old female B6 or NOD mice were cultured for 6 days at 1 × 10$^6$ cells ml$^{-1}$ in complete RPMI medium with 10 µg ml$^{-1}$ CpG-ODN 1018 to allow expansion of memory B cells. Serially diluted CpG pre-activated splenocytes (5 × 10$^4$ to 4 × 10$^5$ cells per well) were then added in the ELISPOT plate. After a 18 h culture, plates were washed three times with PBS/0.25% Tween-20, three times with PBS and then incubated with alkaline phosphatase-anti-mouse-IgG (1:1000; Sigma-Aldrich) diluted in PBS/2% BSA for 2 h at room temperature. Plates were then washed and phosphatase activity measured adding 100 µl per well substrate (Bio-Rad). Reaction was blocked by extensive washing with tap water after 15 min incubation. Spots were counted with an AID camera.

**Murine T-cell studies.** A peptide library of 15-mers overlapping by 12 amino acids covering the whole sequence of mIL-2 (including the signal peptide) was generated (GL-Biochem). Peptides (10 mM) were stored in dimethylsulfoxide (DMSO) at − 20 °C until use. For initial screening, peptides were divided in 17 pools of 6 peptides/each with an overlap of 3 so that the final concentration of each peptide was 3 or 10 µM. A first screening using pools of six peptides (B6: $n = 4$,

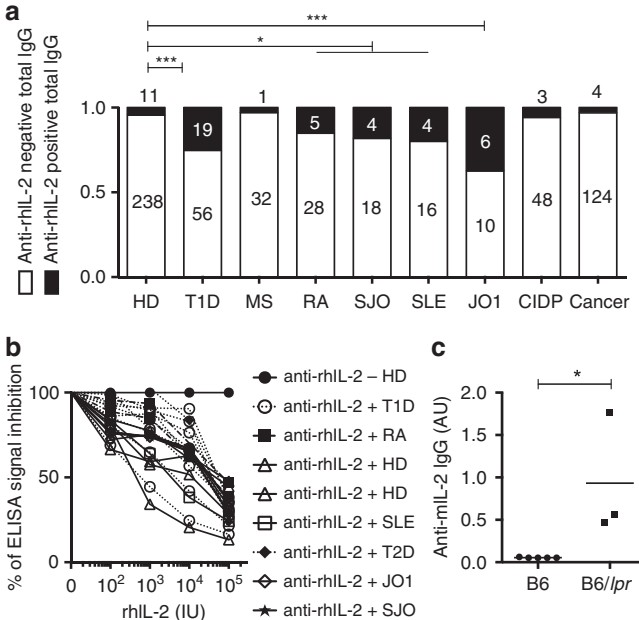

**Figure 6 | High frequencies of anti-IL-2 autoantibodies are present in different autoimmune diseases. (a)** Serum samples were obtained from healthy donors (HD, $n = 249$), T1D ($n = 75$ in the three pooled cohorts), multiple sclerosis (MS; $n = 33$), Sjögren syndrome (SJO; $n = 22$), anti-JO1 positive polymyositis (JO1; $n = 16$), rheumatoid arthritis (RA; $n = 33$), systemic lupus erythematosus (SLE; $n = 20$), chronic inflammatory demyelinating neuropathy (CIPD; $n = 51$) and cancer (Cancer; $n = 128$) patients. Cancer patients were used as controls for a non-autoimmune disease. Left panel: serum titres of anti-rhIL-2 IgG in the different cohorts. Dashed line indicates the threshold of positivity. Right panel: percentage of anti-rhIL-2 positive subjects among the different cohorts. Symbols represent individual subjects and horizontal bars are the medians. *$P < 0.05$; ***$P < 0.001$ (Fisher exact test). **(b)** To control the anti-rhIL-2 autoantibody specificity a competition ELISA was performed. Sera from one anti-rhIL-2-autoantibodies⁻ healthy donor (HD) or from anti-rhIL-2-autoantibodies⁺ samples from HD and T1D, T2D, SLE, RA, Sjögren syndrome (SJO), polymyositis (JO1) patients were pre-incubated or not for 1h with increasing amounts of free recombinant rhIL-2 and titres of anti-rhIL-2 were then quantified by ELISA. Shown is percentage of inhibition of the ELISA signal calculated as (OD with rhIL-2 competition) × 100/(OD without competition). Curves represent individual subjects. Data are cumulative of three independent experiments. **(c)** Titres of anti-mIL-2 IgG were quantified by ELISA in the sera of B6 and 4-month-old lupus-prone mice B6/*lpr* mice. Each serum was tested once or twice in duplicate. Symbols represent individual mice and horizontal bars are the medians. *$P < 0.05$ (non-parametric Mann-Whitney test).

NOD: $n = 4$) allowed the identification of two potentially immunogenic regions. In a different set of mice, pooled peptides that showed a significant response in NOD compared to B6 mice were single-tested at a concentration of 3 and 10 μM. Splenocytes from 10- to 18-week-old female B6 or NOD mice were cultured in triplicate ($4 \times 10^5$ cells/150 μl per well) in X-Vivo 15 serum-free medium (Lonza) containing DMSO (negative control), class II-restricted BDC2.5 mimotope peptide (P31; YVRPLWVRME; 3–10 μmol l⁻¹), aCD3-CD28 beads (positive control, ratio 1bead:1cell, Life Technologies) or peptides. After 72 h, 50 μl of supernatant was saved for the analysis of cytokine production. IFN-γ in culture supernatants were measured with IFN-γ cytometric beads assay (CBA) Flex Set (BD Biosciences) according to the manufacturer's instructions.

**Human serum and plasma samples.** Adult healthy donors, type 2 diabetic or T1D (cohort 1) patients were recruited at the Diabetology Unit of the Pitié Salpétrière Hospital in Paris (France) following the local ethic guidelines. Serum samples from healthy donors and T1D (cohort 2) patients were provided by the DASP program (http://www.cdc.gov/labstandards/diabetes_dasp.html). Healthy donors and T1D patients from cohort 3 were recruited at the San Raffaele Institute in Milan (Italy), following the local ethic guidelines. Only adult T1D patients were included in the final analysis. Healthy donors matched with patients suffering from different inflammatory/autoimmune diseases were recruited at INSERM U905, Rouen (France). For this cohort, patients were classified according to established classification criteria: ACR revised criteria for systemic lupus erythematosus[27] with anti-dsDNA autoantibodies, ARA criteria for rheumatoid arthritis[28] with anti-CCP antibodies and/or rheumatoid factor, revised European criteria for primary Sjögren's syndrome[29] with anti-SSA and/or anti-SSB autoantibodies, Troyanov criteria for overlap myositis with anti-tRNA-synthetase Jo-1 autoantibody[30], and described previously[31]. Patients suffering from MS according to the 2005 McDonald criteria[32] and chronic inflammatory demyelinating polyneuropathy according to EFNS criteria[33], were recruited at Henri Mondor Hospital/UPEC University, Créteil (France). Sera were collected before initiation of

methylpredinisolone in case of relapse of MS and before initiation of intravenous immunoglobulin treatment in case of chronic inflammatory demyelinating polyneuropathy. This retrospective study received ethical standards committee approval, and patients were informed of the collection of their anonymous data for research according to French standards. Sera from patients suffering from different cancers (melanoma, head and neck, lung, colorectal or breast) were obtained from the Centre de Ressources Biologiques at the Curie Institut Paris (Dr S. Saada), in accordance with the Local Ethical Guidelines. Serum samples were conserved at $-20$ or $-80 °C$ until use.

**Quantification of anti-human IL-2 autoantibodies.** Serum titres of anti-rhIL-2-autoantibodies were assessed by ELISA. Microtitre 96-well plates (Medisorp, Nunc) were incubated overnight at $4 °C$ with 100 μl per well of carbonate coating buffer containing $10^5$ IU ml⁻¹ rhIL-2 ('IL-2 coated wells') or buffer alone ('uncoated wells', blank). After blocking with PBS/2% BSA for 2 h, plates were incubated with 50 μl serially diluted serum samples for 2 h at room temperature. After extensive washing with PBS/0.1% Tween 20, HRP-conjugated anti-human IgG (1:2,000; Dako) was added to each well and the plates were kept at room temperature for 1 h. Peroxidase activity was measured with TMB substrate as before. Standard curve was generated using two-fold serial dilutions of rat anti-human IL-2 (clone MQ1-17H12, eBioscience) revealed with an HRP-conjugated goat anti-rat Ig. Arbitrary Units for each sample were calculated using the O.D. value obtained after subtraction of the blank. For the IL-2 competition ELISA, purified IgG from T1D patient sera (at a constant concentration of 50 nM) were incubated in 384-well microtitre plates (Maxisorp, Nunc), coated overnight with 10 μg ml⁻¹ Proleukin or 1 μg ml⁻¹ each H1, H3, and B influenza variants and blocked with 0.5% w/v BSA, in competition with decreasing concentrations of soluble Proleukin (from 9 μM to 5 pM in 3-fold dilution series). Non-competed anti-IL-2 or anti-influenza antibodies were detected with HRP-conjugated Goat anti-human IgG (SouthernBiotech).

**Human B- and T-cell cultures.** PBMCs were separated by Ficoll-Paque PLUS centrifugation (GE Healthcare) from T1D patients and buffy coats obtained from samples from healthy blood donors (Saint Antoine-Crozatier Blood Bank, Paris). Circulating total human B cells were enriched with EasySep kit Stem cell technology (STEMCELL tech. 19054) followed by isolation of class switch memory B cells ($CD20^+IgG^+CD27^+$) by cell sorting with FACSAria II cytometer using the following antibodies: CD27-v450, IgD-PE (clone M-7271 BD), CD20-Alexa488 (Clone 2H7 Biolegend), CD3-CD14-CD16-CD56-PE-Cy5 (Clone UCHT1, RMO52, 3G8, N901 Beckmann Coulter) and IgA-IgM-AlexaFluor647 (Jackson ImmunoResearch). Purified class-switched memory B cells were plated at 4 cells per well in 384-well plates and stimulated based on previously published protocols[18]. After 12 days, supernatants from human B-cell cultures were collected and analysed by ELISA for the presence of anti-IL-2, insulin and influenza IgG antibodies.

To investigate human T-cell reactivity against IL-2, a library of 15-mer peptides with 10 amino acid overlaps covering the whole native hIL-2 was used. We further included four peptides covering regions comprising the two mutations introduced in therapeutic recombinant rhIL-2 (Proleukin) relative to native IL-2, namely an alanine→methionine substitution at the N-terminus and a cysteine→serine replacement at position 145 (125 in the Proleukin sequence). Proleukin sequences were synthesized (GL-Biochem; Supplementary Table 4). Intracellular IA-2 ($10 \mu g\, ml^{-1}$ final concentration; amino acids 214–591, endotoxin $5 EU\, mg^{-1}$; kindly provided by J.F. Elliott, University of Alberta, Edmonton, Canada), adenoviral (AdV) lysate and phytohemagglutinin (PHA) were included as positive controls. Frozen-thawed PBMCs from T1D and healthy subjects recruited in Paris were processed as described[19,34]. PBMCs ($10^6$ per well in 96-well flat-bottom plates) were stimulated for 48 h with peptides using the accelerated co-cultured dendritic cell (acDC) technology[19]. At the end of the 48 h culture, PBMCs were washed and $2 \times 10^5$ cells per well distributed in triplicate wells of 96-well polyvinylidene difluoride ELISPOT plates coated with an IFN-γ antibody. After 6 h of incubation, plates were developed as described[35], counted on a BioSys 5000 Pro-SF Bioreader and results expressed as IFN-γ spot-forming cells (SFC)/$10^6$ PBMCs after background subtraction. For initial screening, peptides were divided in pools of 3/each (2/each for paired native hIL-2 and Proleukin sequences) and used at a final concentration of $5 \mu M$ per each. T-cell responses were scored as positive for SFC counts $> 6$ s.d. above spontaneous background responses in the presence of DMSO diluent alone. Individual peptides from positive pools were further tested ($10 \mu M$ final concentration) and scored as positive for IFN-γ SFC numbers $> 6.5/10^6$ PBMCs after background subtraction (which was $< 10$ SFC/$10^6$ PBMCs in all cases). Cut-offs were selected according to receiver-operator characteristic (ROC) analyses as described[35].

Anti-IgG ELISA was used to determine which wells contained B cells that responded to the stimuli; the response efficiency (20–35% of all seeded B cells) was used to determine the number of BCRs under investigation for each of the single-antigen ELISAs (influenza, IL-2, or insulin). RNA from cell lysates of wells positive for anti-IL-2 IgG (but neither of the other antigens nor a BSA negative control) was reverse transcribed and this cDNA was used to clone the respective heavy and light chain genes in each well into mammalian expression vectors, whereby the $V_H$ and most of the $C_H1$ domain of the heavy chain, and the $V_K$ or $V_L$ domain were grafted onto backbones containing the remaining IgG1, IgK, or IgL C domains, respectively. After eliminating PCR errors, unique identified sequences were re-expressed as heavy/light chain pairs to find the correct sequence pairing reproducing the antibody in the cell culture supernatant. These sequences were input into the IgBLAST database to identify V and J segment usage, as well as to quantify somatic hypermutation relative to the closest germline alignment.

**Statistical analyses.** For diabetes incidence experiments, the number of mice per group was calculated based on our experience[8,36]. For serum transfer experiments, NOD mice were randomized with GraphPad QuickCalcs (www.graphpad.com/quickcalcs/) to receive NOD serum, anti-mIL-2-autoantibodies-depleted NOD serum or S4B6. No randomization method was used for the other experiments performed. Investigators were not blinded throughout the whole study. Correlations were performed using a non-parametric Spearman correlation test. As sample distribution throughout the manuscript was not normal (as determined by a D'Agostino and Pearson omnibus normality test), differences between separate groups or related groups were analysed using a two-sided Mann-Whitney test or a two-sided Wilcoxon matched-pairs signed rank test, respectively, with $P < 0.05$ taken as statistical significance. We used the nonparametric Kruskal-Wallis procedure followed by the Dunn's multiple comparisons test to evaluate the difference in the anti-mIL-2 autoantibody titres among NOD mice according to their age and disease status. When comparing the percentage of anti-rhIL-2-autoantibodies$^+$ patients within two different groups or the percentage of subjects responding to rhIL-2 peptides in two different groups, we used a Fisher's exact test, with $P < 0.05$ taken as statistical significance. For ELISA tests quantifying anti-rhIL-2 autoantibodies, we fixed the threshold of positivity at a value of 24.3AU, which allowed discrimination of healthy donors and T1D subjects with 95% specificity. This cut-off was calculated with an ROC curve with a 95% confidence interval, using the T1D subjects of cohorts 1, 2 and 3 ($n = 75$), as patients; and the healthy donors coming from these cohorts ($n = 103$), as controls. This cut-off value was applied for all the samples. All statistical analyses were performed using GraphPad Prism v6 software.

**Data availability.** The data that support the findings of this study are available from the corresponding author on request. The authors declare that all other data supporting the findings of this study are available within the article and its supplementary information files.

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

## Acknowledgements

L.P. was supported by the Ministère de la Recherche and N.G.N. was supported by a fellowship from Ligue Nationale Contre le Cancer. This work was supported by the ARC N° PJA 20131200444, Agence Nationale de la Recherche (ANR-09-GENO-006-01), EFSD/JDRF/NN 2011 and INSERM/DGOS 2011 (to E.P.) and by Inserm-Transfert and Aviesan/AstraZeneca 'Diabetes and the vessel wall injury' programme (to R.M.), J.Y. and P.S. contribution was funded by the Canadian Institutes of Health Research. P.S. is a Scientist of the Alberta Innovates—Health Solutions and a scholar of the Instituto de Investigaciones Sanitarias Carlos III. The JMDRC is supported by the Canadian Diabetes Association. We particularly thank Silvina Villar (Instituto de Immunologia, Facultad de Ciencas Medicas, Universidad Nacional de Rosario, Rosario, Argentina) for providing B6/*lpr* sera and P. Blancou (INSERM/UMR1080, CNRS/UMR6097, Université de Nice Sophia Antipolis, France) for providing NOR sera and helpful scientific discussion as well as C. Boitard (INSERM/U1016, Cochin Institute, Paris, France) for advise with NOD^Idd6 mice. We thank S. Amigorena and AM. Lennon-Duménil (INSERM U932) for critical reading of the manuscript. We thank the patients, the animal care team members Christelle Enond, Flora Issert, François Bodin, Serban Morosan (all from the Centre d'Exploration Fonctionnelle, Université Pierre et Marie Curie) and Olivier Lebhar, Armelle Halle, Céline Daviaud, Isabelle Grandjean and Virgine Dangles-Marie (all from the Institut Curie, Paris).

## Author contributions

L.P., A.B., A.V., N.N., J.M.L., M.S., X.C.L. and C.S. designed and performed experiments and analysed data. M.-C.G. and R.M. designed, performed and analysed human T-cell experiments. D.L., O.B., A.C., J.L.C., U.C.R., J.Y., P.S., M.B. and A.H. provided serum samples and contributed to scientific discussions. E.P. supervised the project. L.P., E.P., J.M.L. and E.T. wrote the manuscript with input from all coauthors.

## Additional information

**Competing financial interests:** E.P., L.P, and I.C. together with Inserm have filed a provisional patent application that relates to diagnostic methods of an autoimmune disease (WO 2015162124 A1). All other authors declare no competing financial interests.

