## [Peer Review File · Nature Communications]

Reviewers' Comments:

Reviewer #1 (Remarks to the Author)

In the present study Perol and co-workers identified interleukin-2 (IL-2) as a new and non-pancreatic target of autoimmunity in type 1 diabetes and other autoimmune diseases. These findings were first observed in diabetes-prone NOD mice and then confirmed in humans. These observations are novel, and, considering that IL-2 is crucial to induce Treg function and thus maintain tolerance, they provide potentially relevant mechanistic information on the pathogenesis of different autoimmune diseases.

Some aspects of the study, however, require additional clarification:

1. In Suppl. Fig. 1, pre-diabetic NOD mice are treated with 250.000, 500.000 or 1 million IU hIL-2. In the figure, however, there is no clear discrimination between these 3 doses. Were their effects identical, allowing pooling of the data? This should be mentioned in the legend or text. Furthermore, are similar results observed with mIL-2?
2. In Fig. 1C, proliferation of CTLL-2 cells cultured with mIL-2 is expressed as % of the respective controls. Please indicate in the text or legend the absolute control values for NOD and B6.
3. No information is provided on the T2D cohort studied in Fig. 3A (gender? Age? Islet-autoantibody negative? Time from diagnosis? etc).
4. Please define HD (healthy donors) in the legend for Fig. 4.
5. The data obtained in T1D humans are less impressive than in NOD mice (only a minority of patients show anti-IL2 antibodies), reinforcing the concept that the human disease is heterogeneous, which may have impact for therapy. This should be commented upon in the text.

Reviewer #2 (Remarks to the Author)

The manuscript by Perol and colleagues describes an interesting and striking set of observations in which B and T cell reactivity specific for IL-2 is detected in NOD mice and type 1 diabetic patients. It is well established that dysregulation of the IL-2/IL-2 receptor axis is associated with type 1 diabetes (T1D) in both NOD mice and humans. The latter is mostly associated with aberrant expansion and/or function of FOXP3+Treg. Autoimmunity to IL-2 on the other hand may represent a novel mechanism that contributes to the progression of T1D by limiting IL-2 needed to maintain/expand a protective FOXP3+Treg pool, for instance. Overall this is a well written manuscript, the work is comprehensive, and the data provided are convincing. A key strength of the study is demonstration that anti-IL2 Ab are detected in different cohorts of T1D patients, in particular in recent onset individuals (cohort 2).

The study, however, would be markedly enhanced if a pathogenic role for anti-IL2 Ab was demonstrated in NOD mice. Results with the NOD congenic lines show that anti-IL2 Ab are insufficient to drive disease raising the possibility that autoreactivity to IL-2 may simply reflect a "by product" of inefficient self tolerance induction in NOD mice (and T1D patients). In the Materials and Methods a description for serum transfer experiments is included; the results for these experiments however are not provided in the manuscript. Do the undepleted versus depleted sera transfer/exacerbate disease and/or impact the number/frequency of islet resident Foxp3+Treg and conventional T effectors in NOD mice? Are IL-2 peptide-specific T cells detected in the pancreatic lymph nodes and islets of NOD mice (which are more disease relevant tissues than the spleen)? Do NOD APC pulsed with native IL-2 stimulate T cell reactivity?

Minor Comments:

Fig. 2 needs asterisks in the plots to denote the statistical differences.

Supplementary Fig. 2 needs statistical analyses.

As noted above the description in the M&M for the serum transfer experiment should be deleted if the data is not provided.

Reviewer #3 (Remarks to the Author)

In the present manuscript the authors highlight an important novel role of IL-2 as an autoimmune target in murine and human Type 1 diabetes. Specifically, the authors identify anti-IL-2 autoantibodies as well as T cells recognizing a single epitope in IL-2 in the setting of murine and human Type 1 diabetes. While the findings are certainly of relevance and interest, in the view of this reviewer critical questions remain in order to understand the contribution of aberrant immune responses against IL-2 in disease pathogenesis and progression and especially with respect to the discussed impairment in immune tolerance mechanisms.

Specific questions and comments:

1. Where does this IL-2 related autoimmune attack originate? Is there an impact in the thymus since the role of IL-2 for Treg differentiation and survival is particularly critical in the thymus.
2. The causative role of the aberrant immune response against IL-2 is not convincingly shown: Do IL-2 autoreactive T cells transfer disease?
3. Regarding the relevance of IL-2 autoreactivity for progression to overt T1D: Is there an interpretation why those IL-2 related autoantibodies are higher in NOD mice with overt T1D? This is an important question since other autoantibodies, especially IAAs wax and wane in NOD mice and often decline upon overt diabetes development.
4. Is there a correlation of IL-2 related autoantibodies with TFH cells or their precursors circulating in the blood?
5. With respect to their contribution in disease pathogenesis: It would be interesting to see whether IL-2 autoreactive B and especially T cell responses can be identified in more disease-relevant regions such as pancreatic lymph nodes or directly infiltrating the pancreas. This would be of particular interest for the T cell responses directed against IL-2.
6. In Figure 2 c in the splenocytes experiment using the IL-2 related epitope stimulation: The T cell responses against those epitopes appear rather weak with respect to the concentrations used (10 μ M). Are there differences or stronger responses observed when proliferation of T cells is analyzed?
7. The authors indicate that these IL-2 related autoantibodies are present in NOR mice. Does this argue against their causative role for disease progression? Do the T cell responses against IL-2 differ in T cells from NOR mice? Especially with respect to differences in cytokine responses upon stimulation? It would be interesting to look not only for IFN γ but also for anti-inflammatory responses such as IL-10.
8. Do these IL-2/IL-2 autoantibody complexes contribute to the autoimmune attack (accelerating vs. decelerating?)
9. In the human setting: Are these IL-2 related autoantibodies present in the pre-diabetic state,

e.g. in non-diabetic individuals with ongoing islet-autoimmunity? Do these responses correlate with age as suggested in the murine setting? Are the autoreactive T cell responses directed against IL-2 present in non-diabetic islet-autoantibody positive individuals?

Reviewer #4 (Remarks to the Author)

This interesting paper demonstrates that there autoantibodies to IL-2 are present in both the diabetic NOD mouse and human T1D patients. This finding is quite significant as it implies the potential for diminished available circulating IL-2 that is required for Treg survival and function. The data are original but the concept of anti-cytokine antibodies in autoimmunity is well documented. Their data and methodology are state of the art and their conclusions do not overstep the reality of the data. I think this manuscript is fine as is and cannot think of additional experiments that might be required. They reference appropriately and write with clarity.

POINT-BY-POINT REPLY

(our answers are in blue, and we have highlighted in yellow the changes introduced in the revised version of the manuscript)

Reviewers' comments:

Reviewer #1 (Remarks to the Author):

In the present study Perol and co-workers identified interleukin-2 (IL-2) as a new and non-pancreatic target of autoimmunity in type 1 diabetes and other autoimmune diseases. These findings were first observed in diabetes-prone NOD mice and then confirmed in humans. These observations are novel, and, considering that IL-2 is crucial to induce Treg function and thus maintain tolerance, they provide potentially relevant mechanistic information on the pathogenesis of different autoimmune diseases.

Some aspects of the study, however, require additional clarification:

- ✓ In Suppl. Fig. 1, pre-diabetic NOD mice are treated with 250.000, 500.000 or 1 million IU hIL-2. In the figure, however, there is no clear discrimination between these 3 doses. Were their effects identical, allowing pooling of the data? This should be mentioned in the legend or text.

As requested by reviewer 1, we have modified Supplementary figure 1 text, figure and legend, adding the detailed analysis of survival and diabetes incidence among male and female NOD mice treated with PBS, 250000, 500000 or 1000000 UI hIL-2.

Main text modified at page 21 and 22 of the MS

Fig. S1: High-doses hIL-2 injection in NOD induce neutralizing anti-hIL-2 antibodies

Supplementary text for Fig. S1

To analyze if doses 10-, 20-, or 40-fold higher than what we have previously shown to be effective to revert hyperglycemia in new onset diabetic NOD mice could increase treatment efficacy, we administered 250,000, 500,000 or 1 million IU hIL-2 to pre-diabetic NOD mice (36). We observed that these higher hIL-2 doses were, in a dose-dependent manner: i) lethally toxic in half of the mice; ii) precipitated diabetes onset in around 25% of them; or intriguingly, iii) induced no apparent clinical signs in around 25 % of the mice, even after a 30-day administration (Supplementary Fig. 1a-c). Interestingly, after 5 days of treatment, mice responded to all doses of administered hIL-2 by increasing Treg cell frequencies, which returned to pre-treatment levels by the 30th day of IL-2 administration (Supplementary Fig. 1d). We reasoned that mice that survived to 30 days of high-dose hIL-2 treatment had probably developed antibodies capable of neutralizing the injected hIL-2. Indeed, only sera from the surviving hIL-2 treated NOD mice, but not from untreated C57BL/6 (B6) mice demonstrated high titers of hIL-2Abs as detected by ELISA (Supplementary Fig. 1e). Furthermore, those sera efficiently neutralized hIL-2 biological activity in an *in vitro* assay using IL-2-dependent CTLL-2 cells (Supplementary Fig. 1f), suggesting that they were responsible for the *in vivo* resistance to the side effects of high hIL-2 doses.

(a-f) Five-to-14 week-old male or female NOD mice were daily treated with PBS or high-doses hIL-2 (250,000; 500,000 or 1,000,000 IU) over 30 days. (a-b) Kaplan-Meier survival curves of treated female (a, top panel) or male (b, top panel) mice; and diabetes incidence in female (a, bottom panel) or male (b, bottom panel) mice. (c) Percentage of dead, diabetic or alive and non-diabetic NOD mice after 30 days of treatment; IL-2-treated: pool of (250,000; 500,000 and 1,000,000 IU IL-2 treated mice. (d) Percentage of Foxp3⁺ among CD3⁺ CD4⁺ splenocytes of NOD mice treated for 5 to 30 days with high-doses IL-2 or PBS. (e) Serum anti-hIL-2 IgG titers of untreated B6 mice and pre-diabetic NOD mice treated for 0, 7 or 30 days with high-dose IL-2. (f) Proliferation of CTLL-2 cells cultured for 3 days with 3 IU/mL hIL-2 and serially diluted serum from B6 (closed circles) or NOD mice treated for 30 days with high-dose hIL-2 (open circles). Proliferation is expressed as percentage of control (CTLL-2 cultured for 3 days with 3 IU/mL hIL-2 without mouse serum). Data are cumulative of at least two independent experiments. ns, not significant. *** $P < 0.001$ (non-parametric Mann-Whitney test).

✓ **Furthermore, are similar results observed with mIL-2?**

It has been previously established that human IL-2 acts on mouse cells in a comparable way to mouse IL-2. Indeed, the majority of published work on mouse experiments has been done with human IL-2/Proleukin (Ho-Jin Shin, Blood 2011, Dansokho C, brain 2016 ; González FB, Brain Behav Immun. 2015, Pérol L, Immunol Lett. 2014 ; Pilon CB, Am J Transplant. 2014 ; Baeyens , Diabetes. 2013 ; Grinberg-Bleyer Y, J Exp Med. 2010). Moreover, the use of human recombinant IL-2 in pre-clinical mouse models presents the possibility to explore also therapeutic targeting of human IL-2, which could be relevant for both autoimmune as well as cancer studies.

✓ **2. In Fig. 1C, proliferation of CTLL-2 cells cultured with mIL-2 is expressed as % of the respective controls. Please indicate in the text or legend the absolute control values for NOD and B6.**

We have modified the text as requested by the reviewer. Revised version of the MS page 11:

(c) Proliferation of CTLL-2 cells cultured for 3 days with 1 ng/mL mIL-2 and different concentrations of B6 (closed circles) or NOD (open circles) sera. Proliferation is expressed as percentage of control (CTLL-2 cultured for 3 days with 1 ng/mL mIL-2 without mouse serum, mean cpm of 84590).

✓ **3. No information is provided on the T2D cohort studied in Fig. 3A (gender? Age? Islet-autoantibody negative? Time from diagnosis? etc).**

As requested by the reviewer, we have now added in supplementary table 3 all the available demographic and clinical characteristics of the T2D cohort and added in page 6:

... or type 2 diabetic (T2D) patients (who present chronic hyperglycemia in the absence of

islet autoimmunity) (**Fig. 3a-b and Supplementary Tables 2-3**)

Supplementary Table 3 – Demographic and clinical characteristics of the T2D cohort

	T2D Cohort (n=24)
Sex, no. (%) female	10 (41.7)
Age at T2D onset, median (range), years	49 (29, 64)
Age at sample collection, median (range), years	60 (41, 73)
Body mass index (BMI), median (range), kg/m ²	31.1 (18.6, 44)

French clinics do not usually assay for islet autoantibodies in T2D patients who have typical clinical features of metabolic syndrome (as shown in our cohort by an elevated BMI).

✓ **4. Please define HD (healthy donors) in the legend for Fig. 4.**

We have modified the text at page 13 as indicated by the reviewer.

✓ **5. The data obtained in T1D humans are less impressive than in NOD mice (only a minority of patients show anti-IL2 antibodies), reinforcing the concept that the human disease is heterogeneous, which may have impact for therapy. This should be commented upon in the text.**

We thank the reviewer for this comment and agree with their observation. As the reviewer points out, T1D is one of the most heterogeneous autoimmune diseases, and we must also consider the disease kinetics, which in a human setting cannot be as easily controlled as in mouse studies. Moreover, as mentioned on page 7 of the MS, intriguingly, the presence of low levels of IL-2-complexed anti-IL2 IgG in healthy individuals has been described some time ago, indicating that human anti-IL2 responses could have a role in general homeostatic control of the immune responses beyond the disease setting. We have modified the main text of the MS, page 6:

“The reduced penetrance of hIL-2AAb in T1D patients relative to the NOD mice may be attributed to genetic and temporal heterogeneity of the human disease, such that only a subset of diabetic patients are phenotypically similar to the mouse model with respect to anti-IL2 antibodies”

Reviewer #2 (Remarks to the Author):

The manuscript by Perol and colleagues describes an interesting and striking set of observations in which B and T cell reactivity specific for IL-2 is detected in NOD mice and type 1 diabetic patients. It is well established that dysregulation of the IL-2/IL-2 receptor axis is associated with type 1 diabetes (T1D) in both NOD mice and humans. The latter is mostly associated with aberrant expansion and/or function of FOXP3⁺Treg. Autoimmunity to IL-2 on the other hand may represent a novel mechanism that contributes to the progression of T1D by limiting IL-2 needed to maintain/expand a protective FOXP3⁺Treg pool, for instance. Overall this is a well written manuscript, the work is comprehensive, and the data provided are convincing. A key strength of the study is demonstration that anti-IL2 Ab are detected in different cohorts of T1D patients, in particularly in recent onset individuals (cohort 2).

- ✓ 1- The study, however, would be markedly enhanced if a pathogenic role for anti-IL2 Ab was demonstrated in NOD mice. Results with the NOD congenic lines show that anti-IL2 Ab are insufficient to drive disease raising the possibility that autoreactivity to IL-2 may simply reflect a "by product" of inefficient self tolerance induction in NOD mice (and T1D patients).

As the reviewer pointed out, our data indicate that anti-IL-2 autoantibodies and anti-IL-2 T cells *per se*, are insufficient to induce full-blown disease. We agree that the anti-IL2 response is part of a misregulated tolerance process in NOD mice likely taking place in the periphery. Nevertheless, we believe that these findings are novel and relevant not only for a better understanding the pathogenesis of T1D, but also to exploit the role of anti-cytokine antibodies in unrelated pathologies. The causative relationship between autoantibodies and clinical signs is one aspect of the pathogenesis of autoimmunity, but it is not the only one. If we take as an example the first organ-specific autoimmune disease to be described, Hashimoto's thyroiditis, the presence of anti-thyroid antigen autoantibodies is not always correlated with disease: twins may have auto-Abs and not disease, and patients with Hashimoto's thyroiditis may be negative for the auto-Abs. In the particular case of T1D, auto-Abs against insulin have been strongly studied; they have helped to understand the disease pathogenesis, and are used today to diagnose disease, but they do not induce disease upon transfer. In the case of anti-cytokine auto-Abs and disease pathogenesis, although these findings are correlative in nature, their identification has been fundamental for a better understanding of disease pathogenesis, as we discussed in the introduction of our manuscript.

- ✓ 2- In the Materials and Methods a description for serum transfer experiments is included; the results for these experiments however are not provided in the manuscript. Do the undepleted versus depleted sera transfer/exacerbate disease and/or impact the number/frequency of islet resident Foxp3⁺Treg and conventional T effectors in NOD mice?

Since Treg cells are absolutely dependent on IL-2, we reasoned that anti-IL-2 antibodies may impact Treg cell homeostasis by modifying IL-2 bioavailability *in vivo*. To test this hypothesis, we injected NOD mice with serum pooled from mIL-2AAb⁺ mice, depleted or not for mIL-2AAbs (supplementary figure 3 in the revised MS). As a control, we

administered 600 ng of the monoclonal anti-mIL-2 antibody (clone S4B6), which was the estimated amount of mIL-2AAbs present in the injected pooled serum. Injection of this amount of S4B6 antibody did not modify Treg cell proportions (not shown), in accordance with published results showing that Treg cell depletion with S4B6 needs 250 to 1500-fold higher amounts. Thus, we measured CD25 expression, which is highly dependent on IL-2 bioavailability, before and after serum transfer. Compared to mIL-2AAb-depleted serum, undepleted serum, as well as S4B6 administration, induced a reduction of CD25 MFI in blood Treg cells, indicating that mIL-2AAb can contribute to reduced IL-2 availability *in vivo*.

We now added in the MS the following sentence, page 5:

Along these lines, injection of mIL-2AAb-depleted versus undepleted serum in NOD mice induced a reduction of CD25 expression on blood T_{reg} cells, (similar to the injection of equivalent amounts of a control anti-IL-2 neutralizing antibody (S4B6), suggesting that mIL-2AAb could contribute to reduced IL-2 availability and impact T_{reg} fitness *in vivo* (Supplementary Fig. 3).

The results are added as supplementary figure 3:

Supplementary figure 3

An the corresponded legend has been added at page 23

Fig. S3. Effect of anti-mIL-2 autoantibodies on T_{reg} cell homeostasis.

In two independent experiments (#1 and #2), male NOD mice of 12 weeks of age were daily injected for 2 days with total NOD serum ($n=3$), mIL-2AAb depleted NOD serum ($n=3$) or anti-IL-2 (clone S4B6, 300 ng/injection, $n=3$ in experiment #1, $n=2$ in experiment #2). Quantification of CD25 MFI among T_{reg} cells in the blood of mice of the indicated groups. Shown are values obtained in individual mice at day 0, before serum transfer and at day 2, 24 h after the second serum transfer in the two independent experiments. Symbols represent individual mice.

The method section is now modified at page 16 as follows:

Methods:

Serum transfer experiment

Serum was collected from mIL-2A^{high} NOD mice and was pooled. One half of the pooled serum was incubated in mIL-2-coated columns (50 μ g mIL-2 per column, Microlink Protein

Kit, Thermo Scientific) overnight at 4°C with end-over-end mixing. The mIL-2A-depleted fraction was recovered by centrifugation and was submitted to another round of depletion. This procedure led to a 60-fold reduction of the mIL-2A IgG titers, as determined by ELISA (see below for the ELISA protocol). As a positive control, we used an anti-IL-2 monoclonal antibody (clone S4B6, BD Biosciences) at a concentration equivalent to the one estimated in the pooled NOD serum. The concentration of mIL-2A in the pooled serum was estimated by ELISA (see below for the ELISA protocol) using plates coated with mIL-2 (1 μ g/mL) or hIL-2 (1 μ g/mL) and using serially diluted murine anti-human IL-2 IgG (BD Biosciences, clone 5344.111) as a standard. We then daily injected i.p. 12-week-old male NOD mice with 100 μ L of total serum, mIL-2A-depleted serum or S4B6 (3 μ g/mL) during 2 days. Flow cytometry was performed on day 0 and day 2.

To more specifically answer to the reviewer question: “Do the undepleted versus depleted sera transfer/exacerbate disease and/or impact the number/frequency of islet resident Foxp3+Treg and conventional T effectors in NOD mice?”, we could not evaluate the effect of the undepleted versus depleted NOD serum on islet T_{reg} cells because the inter-mouse variability in intra-islet T_{reg} cell proportion and CD25 and Foxp3 expression is high and we can only compare in a reliable way the same parameters pre- and post- serum transfer in the blood.

✓ **3- Are IL-2 peptide-specific T cells detected in the pancreatic lymph nodes and islets of NOD mice (which are more disease relevant tissues than the spleen)?**

Although we agree with the referee that pancreatic lymph nodes and islets of NOD mice are relevant tissues for the disease, we would like to point out findings observed in a recent publication (Wan X et al Journal Experimental Medicine, May 2016), in which the authors show, using an anti-insulin BCR heavy chain transgenic model, that insulin immunoreactivity extends beyond the pancreatic lymph node-islet of Langerhans axis and indicates that circulating insulin can also have a role in activating B cells outside of the pancreas, opening the possibility that also for a “classical” diabetogenic autoantigen such as insulin, other sites could be involved in the generation and maintenance of serum autoantibody levels. Moreover, anti-IL-2 responses are not expected to be tissue-restricted; on the contrary, they could be generated in other peripheral immune organs and subsequently disrupt the generation of Treg and T effector cells that indirectly will also affect the peripheral tissue T cells. Indeed, in a clinical setting, the levels of IL-2-specific memory B cells, anti-IL-2 IgG autoantibodies, as well as autoreactive T cells in circulation could be used as potential biomarkers, to be followed in response to treatment, as we address on page 7 of the manuscript.

✓ **4-Do NOD APC pulsed with native IL-2 stimulate T cell reactivity?**

This question is difficult to address technically, as native IL-2 will stimulate *per se* T cell proliferation. The control group will not be informative: IL-2 is needed for T cell proliferation (needs to be added in the culture).

Minor Comments:

- ✓ **Fig. 2 needs asterisks in the plots to denote the statistical differences.**

We have now corrected the figure, as shown below.

Figure 2

✓ Supplementary Fig. 2 needs statistical analyses.

We have now corrected the figure, as shown below.

Supplementary Fig 2

✓ As noted above the description in the M&M for the serum transfer experiment should be deleted if the data is not provided.

Data has been provided as supplementary figure 3, as detailed above.

Reviewer #3 (Remarks to the Author):

In the present manuscript the authors highlight an important novel role of IL-2 as an autoimmune target in murine and human Type 1 diabetes. Specifically, the authors identify anti-IL-2 autoantibodies as well as T cells recognizing a single epitope in IL-2 in the setting of murine and human Type 1 diabetes. While the findings are certainly of relevance and interest, in the view of this reviewer critical questions remain in order to understand the contribution of aberrant immune responses against IL-2 in disease pathogenesis and progression and especially with respect to the discussed impairment in immune tolerance mechanisms.

Specific questions and comments:

Where does this IL-2 related autoimmune attack originate? Is there an impact in the thymus since the role of IL-2 for Treg differentiation and survival is particularly critical in the thymus.

We thank the referee for addressing this critical question: whether the anti-IL2 response, is due to a central or peripheral tolerance defect or both, and if it has an impact on the thymus.

The possibility that IL-2 autoreactivity is negatively selected in the thymus, similar to pancreas-specific autoantigens is an intriguing hypothesis, but we have to take into account that AIRE specifically induces the expression of non-lymphoid, promiscuous self-antigens. Moreover, a re-analysis of the original data sets of differentially expressed genes in the mTEC of aire-deficient relative to wild type mice did not show IL-2 as significantly different, suggesting that IL-2 expression in mTECs is not under the control of AIRE (Anderson M et al Science 2002). A recent publication identifies *Fezf2* as another self-antigen expression-orchestrating gene in the thymus (Takaba H et al Cell 2015), controlling the expression of a different set of organ specific self-antigens in mTECs. Also in this case, the IL-2 gene is not differentially expressed in *Fezf2*-deficient mice compared to wild type animals.

Whether anti-IL2 responses can have an impact on the thymus *per se*, we note work done in an IL-2 deficient mouse model. IL-2 is not required for aire-dependent thymic clonal deletion of high-avidity diabetogenic clones but it is essential for thymic formation of peripheral regulatory Foxp3-expressing CD4+ T cells (Liston A et al Immune Cell Biology 2007). In NOD mice, we have observed that the percentage of Foxp3+CD4+ T cells is not correlated with the titer of mIL-2AAbs, suggesting that anti-IL2 responses do not affect the overall differentiation of Treg cells. We have modified the manuscript to consider this possibility by adding the following text in page 5:

...Interestingly, while IL-2 deficient mice are defective for thymic formation of a subset of islet-specific regulatory T cells (13), we do not observe a correlation between Foxp3-positive T cell numbers and mIL-2AAb in NOD mice, suggesting this effect is peripheral, and not thymus-driven.

2. The causative role of the aberrant immune response against IL-2 is not convincingly shown: Do IL-2 autoreactive T cells transfer disease?

Ag-specific transfer experiments have mostly been performed with transgenic mice, which are

not available for IL-2 (Berry G J Vove 2013). Moreover, we do not believe that anti-IL2 T cell clones per se are pathogenic. Rather, they participate in an aberrant immune response in NOD mice as well as T1D, and they can indirectly influence other arms of the effector immune system.

- ✓ **3. Regarding the relevance of IL-2 autoreactivity for progression to overt T1D: Is there an interpretation why those IL-2 related autoantibodies are higher in NOD mice with overt T1D? This is an important question since other autoantibodies, especially IAAs wax and wane in NOD mice and often decline upon overt diabetes development.**

Indeed, we have measured IAA antibodies in our NOD colonies, and observed that, as published, IAA are only present in NOD mice and that IAA titers decrease with disease onset (see figure below, panel a). When we studied the correlation of IAA titers and anti-IL-2 titers, we observed a positive correlation only at 14 and 18 weeks but not at 6 or 10 weeks or at diabetes onset (figure 1 for referee perusal).

Figure 1 for reviewer 3 perusal:

(a-b) Sera were obtained at different ages after birth and at disease onset (Onset) in one cohort of female NOD mice ($n=13$). IAA titers were measured as previously published using the same protocol and reagents (Daniel et al., J Exp Med, 2011) and mIL-2AAs were measured as described in the manuscript. (a) Serum IAA titers in NOD mice in function of the age. (b) Correlation between anti-mIL-2 IgG titers at different time after birth and at disease onset (non-parametric Spearman correlation test).

The kinetic of serum autoantibodies is mainly related to the cellular mechanisms by which memory B cells and plasma cells specific for autoantigen are generated. As the reviewer points out, anti-insulin IgG in NOD mice like those in the human, are detectable from the earliest stages of insulinitis (Yu L et al PNAS, 1999) and could be used as a useful early biomarker. During progression from peri- to intra-insulinitis in early diabetic mice, T and B cell infiltration follows a highly regulated process with the formation of lymphoid aggregates characterized by T/B cell segregation, follicular dendritic cell networks, and differentiation of

germinal center B cells, so called tertiary lymphoid structures (TLS) (Astorri E et al Journal Immunology 2010). Thus, serum autoantibodies against insulin could be generated in pancreatic tissue in an early phase and maintained thanks to the chronic exposure to autoantigen, supporting the notion that persistent antigen is crucial for the maintenance of serological memory (Zinkernagel RM Curr Opin Immunol. 2002 Aug;14(4):523-36). In NOD mice with overt T1D where pancreatic tissue as well as insulin as autoantigen are destroyed, the decline of auto-antibodies correlates with a lack of persistent antigen needed to activate memory B cells. On the contrary, in the case of anti-IL-2 responses titers are stable and persistent also in overt T1D where we do not expect a decline of IL-2 levels, since the presence of IL-2 as an autoantigen is less likely to fluctuate with the destruction of pancreatic tissue.

To address this point we have added the following sentence to the manuscript, page 4:

... In contrast to IAA titers, which can oscillate during disease progression in NOD mice (10), mIL-2AAb titers appear at a stable trajectory preceding or concomitant with disease progression. This may be due to the cyclical appearance of insulin as an autoantigen during waves of pancreatic destruction, whereas IL-2 is more persistently present to drive a response.

- ✓ **4. Is there a correlation of IL-2 related autoantibodies with TFH cells or their precursors circulating in the blood?**

Analysis of circulating T follicular helper cells (Tfh) measured as CD4⁺CXCR5⁺CD45RA⁻ (Morita R et al Immunity 2011) shows a significant expansion of Tfh in T1D patients compared to age-matched healthy controls (figure 2 for referee perusal). Indeed, this observation is in agreement with data reported in a previous publication from Kenefick R et al (Journal of Clinical Investigation, January 2015). In the cited manuscript, the authors linked the expansion of circulating Tfh to an impaired IL-2 response with a consequent up-regulation of CXCR5 expression. Stratifying our T1D cohort according to weak, intermediate, and strong IL-2 IgG binders as depicted in figure 3 of our MS, we could not find any correlation between the expansion of Tfh and presence of serum anti-IL2 IgG.

[Figures redacted]

- ✓ **5. With respect to their contribution in disease pathogenesis: It would be interesting to see whether IL-2 autoreactive B and especially T cell responses can be identified in more disease-relevant regions such as pancreatic lymph nodes or directly infiltrating the pancreas. This would be of particular interest for the T cell responses directed against IL-2.**

Although we agree with the referee that pancreatic lymph nodes and islets of NOD mice are relevant tissues for the disease, we would like to point out the intriguing findings observed in a recent publication (Wan X et al Journal Experimental Medicine, May 2016), in which the authors show using an anti-insulin BCR heavy chain transgenic model that insulin

immunoreactivity extends beyond the pancreatic lymph nodes-islet of Langerhans axis and indicates that circulating insulin can also have a role in activating B cells outside of the pancreas, opening the possibility that also for a “classical” diabetogenic autoantigen such as insulin, other sites could be involved in the generation and maintenance of serum auto-antibody levels. Moreover, anti-IL-2 responses are not expected to be tissue restricted; on the contrary, they could be generated in other peripheral immune organs and subsequently disrupt the generation of Treg and T effector cells that indirectly will also affect the peripheral tissue T cells. Indeed in a clinical setting the levels of anti-IL2 specific memory B cells, anti-IL-2 IgG autoantibodies, as well as autoreactive T cells in circulation could be used as potential biomarkers, to be followed in response to treatment, as we address on page 7 of the manuscript.

- ✓ **6. In Figure 2 c in the splenocytes experiment using the IL-2 related epitope stimulation: The T cell responses against those epitopes appear rather weak with respect to the concentrations used (10 uM).**

In fact, as observed in Figure 2c, the amount of IFN-g released is in the same range as that observed for the P31 BDC2.5 epitope, which is known to be highly immunodominant.

- ✓ **Are there differences or stronger responses observed when proliferation of T cells is analyzed?**

We have indeed evaluated proliferation in the above-mentioned assay, but results are less clear than those obtained for IFN-g production (this problem is well-documented in the field of identifying Ag-specific T-cell responses in human and mouse). The favorable read out to evaluate memory effector function of both CD4 as well as CD8 T cells is cytokine secretion and this is the major reason why we decided to test IFN γ secretion (Scotto M, Diabetes, 2012). Moreover, in order to test T-cell reactivity, we have to use total splenocytes in which antigen presenting cells, such B cells, monocytes, and DCs are present and can load the peptides on MHC molecules and lead to the activation of T cells. Proliferation measurement with thymidine incorporation on a heterogenous population could lead to difficulty in interpreting the results due also to the different response kinetics of naive/memory T cells.

- ✓ **7. The authors indicate that these IL-2 related autoantibodies are present in NOR mice. Does this argue against their causative role for disease progression? Do the T cell responses against IL-2 differ in T cells from NOR mice? Especially with respect to differences in cytokine responses upon stimulation? It would be interesting to look not only for IFN γ but also for anti-inflammatory responses such as IL-10.**

The following sentence already present in the manuscript explains our shared view with the reviewer that the anti-IL2 AAb are not sufficient to cause disease: “In these two strains, although insulinitis and diabetes are reduced or absent, mIL-2AAb are present, indicating that, while their presence is associated with T1D development, they are not sufficient to induce T1D”.

- ✓ **8. Do these IL-2/IL-2 autoantibody complexes contribute to the autoimmune**

attack (accelerating vs. decelerating?)

This question is actually the subject of a follow-up study in our labs.

[Figure redacted]

- ✓ **9. In the human setting: Are these IL-2 related autoantibodies present in the pre-diabetic state, e.g. in non-diabetic individuals with ongoing islet-autoimmunity? Do these responses correlate with age as suggested in the murine setting?**

We agree with the reviewer that this is a very important question, and indeed, we have started to evaluate it.

[Figure text redacted]

Do these responses correlate with age as suggested in the murine setting?

Indeed, following the reviewer's comment, we addressed this question within our T1D cohorts and effectively, we observed a mild, yet significant positive correlation between the age of the patients and the titers of hIL-2AAb. We have now added this data as Supplementary fig 4.

Supplementary Figure 4

Legend of Fig. S4, added at page 23 of the manuscript

Fig. S4. hIL-2AAb titers positively correlate with T1D patients' age

Serum samples were obtained from T1D patients and titers of anti-hIL-2 IgG were determined by ELISA as described above. Correlation between anti-hIL-2 IgG titers and age at blood withdrawal in the anti-hIL-2⁺ T1D patients (non-parametric Spearman correlation test).

And amended the text, page 6 as follows:

Interestingly, as in NOD mice, there was a positive correlation between the hIL-2AAb titers and patient age (Supplementary figure 4), but....

- ✓ **Are the autoreactive T cell responses directed against IL-2 present in non-diabetic islet-autoantibody positive individuals?**

We agree with the reviewer that this is a very interesting question that is actually worth a dedicated research program, which will be performed as follow-up of this project.

Reviewer #4 (Remarks to the Author):

This interesting paper demonstrates that there autoantibodies to IL-2 are present in both the diabetic NOD mouse and human T1D patients. This finding is quite significant as it implies the potential for diminished available circulating IL-2 that is required for Treg survival and function. The data are original but the concept of anti-cytokine antibodies in autoimmunity is well documented. Their data and methodology are state of the art and their conclusions do not overstep the reality of the data. I think this manuscript is fine as is and cannot think of additional experiments that might be required. They reference appropriately and write with clarity.

No actions needed.

General comment to the editor and reviewers:

Concerning the raised point “*we strongly encourage you to provide direct evidence that anti-IL2 autoantibodies, or IL-2 autoreactive T cells, functionally contribute to the disease*“, we would like to share with you our view and interpretation concerning this issue. **Our data indicate that anti-IL-2 autoantibodies and anti-IL-2 T cells *per se*, are insufficient to induce full-blown disease.** However, we would like to discuss with you the current knowledge that causal relationships of autoantibodies in general, and anti-cytokine autoantibodies in particular with disease pathogenesis are extremely rare:

- if we take as example the first organ-specific autoimmune disease that was ever described, Hashimoto’s thyroiditis, the presence of anti-thyroid antigen autoantibodies is not always correlated with disease: twins may have auto-Abs and not disease, and patients with Hashimoto’s thyroiditis may be negative for the autoAbs. Also, in the particular case of T1D, *auto-Abs against insulin have been strongly studied, and they have helped to understand the disease pathogenesis, are used today to diagnose disease, but, they do not induce disease upon transfer.*

- in the case of anti-cytokine autoAbs and disease pathogenesis, although findings are correlative in nature, their identification has been fundamental for a better understanding of disease pathogenesis, as we discussed in the introduction of our manuscript.

That said, you will be able to appreciate that our findings are extremely relevant to disease and do have an important impact on health care. Indeed, our message is by no means to say that there is a direct causal relationship of anti-IL-2 autoantibodies and T1D. The message of our paper is that for the first time we describe that in T1D patients, autoimmunity is not only directed to pancreatic antigens, but also to IL-2. This observation, well appreciated by the reviewers, constitutes completely new data, which reinforce the link of T1D physiopathology to multiple defects in the IL-2/IL-2R pathway, further reinforcing the link between impaired IL-2 bioavailability and T1D.

Our observation will change the way we apprehend autoimmunity in T1D, by incorporating the notion that autoimmunity extends beyond tissue antigens or DNA, to cytokines, like IL-2, which is the main regulator of immune tolerance. Moreover, now that low-dose IL-2 therapy is being proposed to T1D patients, quantifying anti-IL-2 autoantibodies will help understand IL-2 therapeutic effects and optimize IL-2 dosing. Finally autoreactivity to IL-2 seems to be a trait common to different autoimmune diseases, making our data even more relevant.

On the other hand, Setoguchi et al have already shown that anti-IL-2 antibodies administered to newborn NOD mice accelerate diabetes onset. We agree with the reviewers that the ideal experiment will be to transfer sera or, even better, purified anti-IL-2 autoAbs to NOD mice to pre-diabetic NOD mice and to observe T1D development. **This is technically not feasible.** To neutralize IL-2 in Setoguchi's paper, NOD mice were injected i.p. with **1 mg** of purified anti-IL-2 on days 10 and 20 after birth. The best that we could do was the experiment now shown in the new figure above in which we transferred serum from NOD mice depleted or not from anti-IL-2 Abs to other NOD mice. The auto-Abs are present in low amounts, and to do this experiment we had to bleed many many mice and the final concentration of the Ab is much much lower than that used in the Setoguchi paper, which are supra-physiologic (it is impossible to inject such a volume of sera to a mouse, and it is technically not possible to obtain such huge amounts of sera from NOD mice to purify such amounts of Abs). As explained in the article, when, as control, we injected the anti-IL2 Ab used by the Setoguchi experiment, we observed a similar effect on the Treg phenotype.

Reviewers' Comments:

Reviewer #1 (Remarks to the Author)

The Authors have answered in an adequate way my critiques.

Reviewer #2 (Remarks to the Author)

The authors have addressed most of the concerns raised in the previous critique. However, lack of any data providing insight into the pathogenicity of anti-IL-2 Ab or IL-2-specific T cells limits the overall impact of the study. At a minimum, demonstration that IL-2-specific T cells are found in disease-relevant draining pancreatic lymph nodes (PLN) and the islets would suggest a pathogenic role even if, as the authors note, IL-2-specific T cells are/may be not "tissue-specific".

Reviewer #3 (Remarks to the Author)

In the view of this reviewer the manuscript has been significantly improved following the revision.

This improvement was achieved by integrating the reviewer's comments into the manuscript, presenting novel data, providing additional figures for reviewer's perusal and complementing the results as well as discussion section of the manuscript.

In the opinion of this reviewer the current version of the manuscript is now acceptable for publication in Nature Communications.